# A Review of Rotational Seismology Area of Interest from a Recording and Rotational Sensors Point of View

**DOI:** 10.3390/s24217003

**Published:** 2024-10-31

**Authors:** Anna T. Kurzych, Leszek R. Jaroszewicz

**Affiliations:** 1Institute of Applied Physics, Military University of Technology, 00-908 Warsaw, Poland; leszek.jaroszewicz@wat.edu.pl; 2Elproma Elektronika Ltd., 05-152 Czosnow, Poland

**Keywords:** sensors, remote sensing, rotational seismology, earthquakes, artificial explosion, civil engineering, structural health monitoring, rotational seismometer

## Abstract

This article reviews rotational seismology, considering different areas of interest, as well as measuring devices used for rotational events investigations. After a short theoretical description defining the fundamental parameters, the authors summarized data published in the literature in areas such as the indirect numerical investigation of rotational effects, rotation measured during earthquakes, teleseismic wave investigation, rotation induced by artificial explosions, and mining activity. The fundamental data on the measured rotation parameters and devices used for the recording are summarized and compared for the above areas. In the section on recording the rotational effects associated with artificial explosions and mining activities, the authors included results recorded by a rotational seismograph of their construction—FOSREM (fibre-optic system for rotational events and phenomena monitoring). FOSREM has a broad range of capabilities to measure rotation rates, from several dozen nrad/s to 10 rad/. It can be controlled remotely and operated autonomously for a long time. It is a useful tool for systematic seismological investigations in various places. The report concludes with a short discussion of the importance of rotational seismology and the great need to obtain experimental data in this field.

## 1. Introduction

The full description of the seismic wave motion, in addition to translational components, should also include three rotational components [1]. The common measurement of the translational motion along three axes and the rotational motion around them can improve S-wave identification. Moreover, such measurements can act as a “point seismic array” to determine incoming waves’ speed, direction, and phase [2]. Therefore, it was necessary to take steps to determine the nature of the rotational effects during earthquakes. For a long time, an analysis of rotational seismic events was carried out only at the theoretical level because no tools allowed for the registration of the rotational speed of soil particles, which often did not exceed fractions of μrad/s. However, as a field of great interest, rotational seismology drove the appearance of instruments enabling the direct measurement of the rotational components of seismic vibrations. Rotational seismology was defined in 2009 as a dynamically evolving field of science covering all issues of rotational movements of the ground originating from earthquakes, explosions, or atmospheric disturbances [3].The attractiveness of this field is confirmed by its wide range of interests, such as strong motion seismology, broadband seismology, earthquake engineering, earthquake physics, seismic instrumentation, and seismic hazards. Physicists who study gravitational waves in ground-based observatories are also interested in this field. Recently, in the global seismology of earthquakes and dynamically induced seismicity, the monitoring of rotational vibrations using new generation meters, i.e., rotational sensors, has been developing [4]. Seismic observations of waves conducted at surface stations prove the occurrence of additional significant effects of rotational vibrations in the wave field [5,6]. Rotational seismology is a promising field focusing on various goals, the achievement of which depends on future research and the development of sensors. The rapid growth of this field is characterized by the appearance of dedicated sections at conferences, an increase in the number of publications in scientific journals, and the emergence of new instruments. Modern acquisition technologies, such as fibre-optic or ring laser gyroscopes, enable the observation of rotational movements and integrate them with records of translational ground movements as six-degree-of-freedom (6DoF) sensors [1]. Seismologists often wonder what the potential applications of rotational seismology are and what benefits it can bring.

This paper aims to review the main recordings in rotational seismology, future research prospects, and seismic tools used in this area. After some essential background preliminary information, the main results of indirect rotation research are first reviewed. Then, the second method of obtaining data is presented in detail. Rotational effect recordings are presented and divided into specific areas of rotational seismology interest: rotational motions generated by natural earthquakes, teleseismic waves investigation, rotation associated with artificial explosions, rotational effects in the mining activity region, as well as the engineering area of rotational seismology. The basic working principle of the rotational sensors used in the presented recordings is explained, along with the main performance parameters and limitations. In sections concerning recording rotational effects connected with artificial explosions and mining activity, the authors included data recorded by a rotational seismograph of their construction—FOSREM (fibre-optic system for rotational events and phenomena monitoring). The presented sensor uses the Sagnac effect to detect the rotational motion component perpendicular to a specific sensor loop. FOSREM is characterized by the ability to measure the rotation rate in broad ranges, both in amplitudes from several dozen nrad/s up to 10 rad/s, and in frequencies up to 200 Hz. It can be remotely controlled and operated autonomously for long periods, making it useful for systematic seismological studies at various locations. The report ends with a brief discussion of rotational seismology’s significance and the necessity of obtaining data.

## 2. Theoretical Studies of Rotational Effects During Earthquakes

An earthquake releases stored elastic energy due to a sudden fracture and movement of rocks inside the Earth. The earthquake type depends on the region where it occurs and the geological makeup of that region. Due to earthquakes’ genesis, they can be divided into volcanic (accompanied by volcanic eruptions, with the Earth shaking preceding the eruption), collapse (the result of the collapse of cave ceilings, mine activities), and tectonic (associated with movements of the lithosphere, orogenic processes). More than a million earthquakes occur per year worldwide [7] and about 150,000 of them are strong enough to be felt [8]. The energy released during an earthquake propagates in a form of seismic waves, divided classically into two basic types: body and surface waves [9]. Body waves propagate from the epicentre in all directions; among them, longitudinal P-waves (primary waves) and transverse S-waves (secondary waves) can be distinguished [10]. P-waves cause changes in the volume of the medium through which they pass by their compression and rarefaction. The particles of the medium vibrate along the direction of wave propagation. However, in the case of the S-wave, the particles of the medium vibrate perpendicularly to the direction of wave propagation. Hence, their twofold polarization is possible: vertical and horizontal. The P-wave velocity is higher than that of the S-wave. In the case of the same propagation medium, the velocity value is about 1.8 times higher. Surface waves spread over the Earth’s surface from the epicentre at a velocity of 3 to 3.8 km/s [10]. They consist of two types of waves, known as Rayleigh and Love waves [10]. The first is a gravitational-type wave, in which the movement of particles occurs along an ellipse set vertically to the direction of the wave. The Love wave is a surface transverse wave with horizontal polarization, causing horizontal vibrations perpendicular to the direction of wave propagation.

The above classical approach to seismology distinguishes only the linear types of vibrations, which differ in polarization, velocity, and vibration direction. The history of developing ground vibration displacement measurements using seismographs began in the mid-18th century. In subsequent years, they have brought a dynamic development of translational seismic measurements by mechanical seismographs, both analogue and digital, and meters of displacement, velocity, and acceleration. Nevertheless, observations of unusual rotational and even helical deformations, which occurred after earthquakes in various forms of architecture, suggested to scientists a different type of particle vibration than linear ones. Kozák cites a summary of historical examples of observed rotational effects during earthquakes [11]. Figure 1a shows examples of pillars, capitals, and tombstones rotationally deformed by the 2009 L’Aquila (central Italy) earthquake [12]. It should also be underlined that rotational components can also play a significant role in the damaging of high-rise buildings; soil–structure interaction effects should be taken into account (Figure 1b,c). In the case of low frequency content of rotation motion, the base of the structure can rotate with an overturning motion (Figure 1c) [13].

The physical description of the rotational effects of earthquakes was based on two models [15]. The description in 1846 by Mallet [16,17,18,19] is cited in seismology as the first model explaining rotational effects during earthquakes. Mallet has used the principles of classical mechanics to explain the observed rotational effects appearing as a phenomenon in the near-field or even at the earthquake’s epicentre. According to Kozák [11], the historical approach is primarily associated with the observed effects on vertically positioned objects composed of blocks or layers separated by horizontal planes that can rotate due to friction. Mallet’s work gives two primary mechanisms of rotational seismic effects, designated by Kozák [15] as Rot1 and Rot2. The Rot1 model indicates that if a body lying on a horizontal plane is subjected to the influence of a translational wave in the horizontal direction, it may rotate if the vertical projection of the centre of gravity onto the contact plane is not identical with the most vital point of adhesion of this body to the support [11].

On the other hand, the mechanism marked as Rot2 indicates that the interactions of the seismic wave components may gradually change the horizontal position of the body, especially when successive wave components propagate to the considered point at a different angle. As a result, a solid fixed on the surface can be gradually rotated around an axis perpendicular to the horizontal plane by individual components of seismic waves [20]. Since the rotational effects were treated as a side effect or even a marginal effect of the earthquake, until the middle of the last century, the explanation formulated by Mallet was entirely satisfactory and often presented by seismologists.

Contrary to the first theoretical models, which treat rotational effects as secondary and marginal phenomena related to the interaction of seismic waves, the second class of models assumes the existence of an independent rotational component. The second-class models have a more profound theoretical approach, containing elements of elastic wave propagation [4,21] resulting from the progress of theoretical research in the field of micromorphic and asymmetric theories of continuum mechanics and physics of non-linear phenomena. Generally, according to [15], this class of models includes mechanisms marked as Rot3–Rot6. The Rot3-Rot5 models are based on linear physics and describe rotational effects as the propagation of elastic waves in an elastic or quasi-elastic medium. The Rot6 model is based on the physics of non-linear phenomena that may occur in an elastic medium under certain specific conditions [4].

The Rot3 model is similar to the Rot1 and Rot2 models. They are all based on the mechanical principles of wave propagation and deal with rotation in the near field or layers below the recording station. Rot3 assumes that the composition, structure of the medium, depth of the local zone, and the parameters characterizing the tectonic stress can cause a specific seismogenic motion [20].

The Rot4 mechanism is related to the actual rotational deformations and the specific properties of the medium, such as micromorphic (where grain rotation and deformation occur) or micropolar (where only grain rotation occurs) mediums, through which seismic waves propagate. Based on the advanced theory of Poulos [22], Moriya [23], and Teisseyre [5], it was theoretically proved that it is possible to generate and then detect the rotational components of seismic waves because of interactions with the medium in which these waves propagate. In addition, the above authors claim that these waves can appear and be recorded at small distances and probably at greater distances from the epicentre. The issue of rotation propagation from the source over long distances seems to be positively resolved in [24,25,26], where micropolar theory yields asymmetric seismic moment tensors that allow a momentum exchange between the rotational microstructure in the source region and the rest of the Earth and thus explicitly accounts for microstructural rupture processes, which naturally engages a macroscopic displacement response.

The Rot5 model was formulated by Roman Teisseyre [4,27,28] and refers to rotation and twisting movements existing in a homogeneous elastic medium. Starting from the theory of an elastic medium with defects, this model is based on additional connections between the asymmetric part of the stresses and the density of the self-rotating nucleus, where the antisymmetric stresses correspond to the stress moments. Thus, this theory proves that rotational components propagating as waves called seismic rotational waves (SRW) can exist even in a uniformly elastic medium [29,30,31].

The subject matter of rotational effects has become the interest of many research groups, leading to the establishment of an International Working Group on Rotational Seismology (IWGoRS). The first IWGoRS workshop occurred in 2007 and is held successively every three years. The upcoming IWGoRS workshop will take place in Poland in 2025 and will be organized by the Opole University of Technology, Opole, Poland, and the Military University of Technology, Warsaw, Poland. One of the most significant and one-of-a-kind projects of IWGoRS took place in the Geophysical Observatory Fürstenfeldbruck, Germany, from 18–22 November 2019. A unique experiment has been organized as “Rotation and strain in Seismology: A comparative Sensor Test” by Felix Bernauer (Department of Earth and Environmental Sciences, LUM, Munich, Germany) and Stefanie Donner (Federal Institute for Geosciences and Natural Resource, Hannover, Germany) [32]. It has gathered more than 40 different rotational motion and strain sensors in one field test. Moreover, the definition of rotational seismology emerged in 2009 as an appealing research domain of all aspects of the Earth’s rotation caused by earthquakes, explosions, and environmental vibrations [33]. This field of study is still developing, and in this paper, the authors included a review of the most essential recordings associated with rotational seismology. The attractiveness of this field is confirmed by its wide range of interests covering two borderline areas of scientific research:

(a) the scope related to geophysical sciences, such as broadband seismology [34], weak and strong seismology [35], earthquake physics [36,37], prediction of seismic hazards [38], research in the seismotectonic field [39], geodesy [40], research on the existence of gravitational waves [41]. In brief, these areas are seismological applications.

(b) the scope related to engineering aspects of earthquakes: behaviour of irregular engineering structures [42,43] and Earth movements caused by the exploitation of mineral deposits, e.g., rock masses [44]. In brief, these are engineering applications.

Nowadays, the analysis of translational and rotational motion, using 6DoF measurements as mentioned in the first paragraph, is gaining momentum [1,45,46,47]. Rotational seismology is the most appealing field in seismology. Nevertheless, much work is needed to build principles and extract as much information from recordings as possible. The benefits of machine learning should also be incorporated into rotational seismology, which can significantly contribute to the development of earthquake catalogues, seismic analysis, ground motion forecasting, and application to geodetic data [48,49,50]. This paper expects the review of rotational event recordings to present an essential improvement in the practical directions of rotational seismology through reliable rotational sensor construction and recorded data.

### 2.1. Mathematical Description

The classical elastic waves in solid bodies can be described using the essential linear theory of elasticity as follows [51]:

Three equilibrium conditions:(1)$$\sigma_{ij,j}+b_{i}=0,$$
kinematic relations:(2)$$\varepsilon_{ij}=\frac{1}{2}\left(u_{i,j}+u_{j,i}\right),$$
with the compatibility constraints:(3)$$\varepsilon_{ij,kl}+\varepsilon_{kl,ij}=\varepsilon_{ik,jl}+\varepsilon_{jl,ik},$$ and the constitutive law:(4)$$\sigma_{ij,j}=\lambda \delta_{ij}\varepsilon_{kk}+2µ\varepsilon_{ij},$$
where *σ_ij_*—the components of the Cartesian stress tensor; *ε_i_*—tensor of the principal strain; $\delta_{ij}$—the Kronecker delta, *u*—the displacement vector of medium particles, *μ*, *λ*—Lame’s elastic constants, which describe the linear stress–strain relationship in an isotropic material medium. The *μ* is called the modulus of transverse stiffness or Kirchhoff modulus and is a measure of the material medium’s resistance to shear. The shorthand notation has been used above (i.e., the subscript *i* is understood to take the sequential values 1, 2, 3; in case of nine quantities, there is a double-subscripted notation *ij* employed, where *i* and *j* range from 1 to 3 in turn; these nine components with a higher form of a vector is called a tensor; an exception is made when two subscripts are identical, such as *kk*; in this case, the Einstein summation convention states that a subscript appearing twice is summed from 1 to 3; partial differentiation is abbreviated using the comma convention).

The above equations must be supplemented with appropriate boundary conditions (kinematic and static) and initial conditions. Therefore, one should focus on isotropic bodies, which constitute the subject of most problems in the linear theory of elasticity. The following transformations can reduce the basic set of differential and algebraic equations of the linear theory:-in the equilibrium equations, one expresses stresses by strains in accordance with physical relationships;-in the obtained equations, one expresses deformations by displacements in accordance with geometric relations. Ultimately, one obtains the differential equation of elastic vibrations in an isotropic medium, i.e., for constant Lame parameters, the index notation takes the following form [52]:
(5)$$\mu u_{i,jj}+\left(\mu +\lambda \right)u_{j,ji}+b_{i}=\rho {\ddot{u}}_{i},$$
where *b_i_*—body force in *i* direction; $\ddot{u}$—acceleration of particles of the medium. In vector notation, Equation (5) takes the following form [52]:(6)$$\left(\mu +\lambda \right)\nabla \left(\nabla \cdot u\right)+\mu \nabla^{2}u+b=\rho \ddot{u},$$
where *ρ*—mass density. A very often used assumption that significantly simplifies the solution is neglecting the contribution of mass forces (*b* = 0). This results in homogeneous equations that are much easier to solve. With this assumption, the influence of external mass forces is often replaced by a statically equivalent system of external surface forces applied appropriately to the body surface. Neglecting mass forces, dividing by *ρ*, and substituting the expressions for the square of the P-wave velocity (*C_P_*^2^ = (*λ* + 2*μ*)/*ρ*) and the square of the S-wave velocity (*C_S_*^2^ = *μ*/*ρ*), one obtains the basic seismic wave equation for a homogeneous medium [53]:(7)$$C_{P}^{2}\nabla \left(\nabla \cdot u\right)-C_{S}^{2}\nabla \times \left(\nabla \times u\right)=\ddot{u}$$

The above equation shows that the propagation speed *C_P_* occurs with divergence ∇·u, which refers to changes in volume or radial displacement (P-wave). The propagation speed *C_S_* is related to ∇×u, that is, to the changes in rotation or lateral displacement (S-wave). This means that, in addition to the registration of translational vibrations, we can also expect the registration of rotational vibrations.

It should be underlined that in the mathematical description, there is a clear correlation between the angular velocity of ground vibrations and the appropriate vector of the time derivative of the displacement of ground vibrations [54]:(8)$$\omega =\left(\omega_{x},\omega_{y},\omega_{z}\right)=\frac{1}{2}\nabla \times \dot{u}=\frac{1}{2}\nabla \times V$$
where ω—angular velocity vector, $\dot{u}$—time derivative of the vibration displacement vector, and V—translational vibration velocity vector.

The importance of the above equations and theories is essential to perform simulations and conversion methods; they provide the three rotational components indirectly, which are presented in Section 3.

### 2.2. The Formalism of Rotational Motion Measurements

A wide range of signal amplitudes characterizes measurements in seismology. The lowest values of measurable amplitudes are determined by natural background noise, which is highly frequency dependent. The largest ground displacements generated by seismic waves reach 1 m, while the lowest value is usually a 1 nm displacement for 1 Hz [55]. It gives a dynamic range of the order of 90 dB. Moreover, the frequency band of interest is wide and depends on the seismic sources, from μHz for Earth tides to 1000 Hz for P- and S-waves. Most seismological instruments measuring translational ground motion are pendulum seismometers and accelerometers [56]. The seismic signals recorded by the sensors are converted into a digital format by analogue-to-digital converters. Typically, the amplitude resolution is one μV with a sampling rate of 100 samples/s. The converted signal is sent to a recorder, usually connected to a computer that collects data continuously or records seismic events only. Recorders and sensors are installed in seismological stations, usually located in remote areas away from human activity. It also causes relatively tricky access. Seismological stations form a seismic network that primarily aims to locate earthquakes and determine their magnitude. Seismic networks range from tiny ones, such as mining networks that record microearthquakes, to global networks that record data from all over the world. Ground movement is generally measured in the *X*, *Y*, and *Z* implementation. Until recently, only three degrees of freedom related to translational motion or acceleration along the Cartesian frame of reference have been recorded. Three additional rotational degrees of freedom during ground vibration measurements can provide new information valuable to the seismological society [34,57,58,59]. As was mentioned previously, they can help understand the internal structure of the Earth’s seismic sources and are also crucial for engineering [60,61], e.g., monitoring of skyscrapers or wind farms. In addition, studies of nature close to ground motion are essential for seismic engineers interested in seismically safe designs and for seismologists studying the physical processes leading to the complexity of ground motion [62].

The basic seismometric parameters used to characterize the intensity of the impact of vibrations on the surface (on buildings, infrastructure, and people) include the following:peak value of displacement, velocity, and acceleration of vertical ground vibrations: *PGD_Z_*, *PGV_Z_*, and *PGA_Z_* [63,64];peak value of displacement, velocity, and acceleration of horizontal ground vibrations in a particular direction: *PGD_H_*, *PGV_H_*, and *PGA_H_* [64,65];the maximum value of the velocity of horizontal ground vibrations *PGV_Hmax_* and the maximum value of horizontal accelerations *PGA_Hmin_* determined as the resultant of the horizontal maximum of the vector length [65,66];duration of the horizontal component of vibration velocity *t_Hv_* and vibration acceleration *t_Ha_* [67];vibration frequency [68,69],response spectrum [64,70];quotient of the peak vibration values *PGA/PGV* [34,71];Arias intensity [72];accumulated absolute speed value *CAV* [68,69];cumulative absolute displacement value *CAD* [69];parameters characterizing rotational vibrations [54,73].

Some definitions and parameters must be presented and described to stay consistent in the paper. Figure 2 shows the axes in the Cartesian coordinate system for translational velocity components (*V_x_*, *V_y_*, *V_z_*) measured by the classical kind of seismometers applied in seismology and components of rotation (*ω_x_*, *ω_y_*, *ω_z_*) measured by rotational sensors. It is the view of the full six-component of seismic ground motion system. The basic parameters describing rotational vibrations directly following from Equation (8) are as follows [54,74]:component of the angular velocity of ground vibrations around the vertical axis *Z*:
(9)$$\omega_{z}=\frac{1}{2}\left(\frac{\partial u_{X}}{\partial Y}-\frac{\partial u_{Y}}{\partial X}\right),$$component of the angular velocity of ground vibrations around the horizontal *Y* and *X* axes:
(10)$$\omega_{Y}=\frac{1}{2}\left(\frac{\partial u_{X}}{\partial Z}-\frac{\partial u_{Z}}{\partial X}\right),$$
(11)$$\omega_{X}=\frac{1}{2}\left(\frac{\partial u_{Y}}{\partial Z}-\frac{\partial u_{Z}}{\partial Y}\right).$$

The vertical rotation rate *ω_z_* is often called yaw, twist, or torsion. The horizontal rotation angles *ω_x_* and *ω_y_* are often called rocking, roll (around the *X*-axis), and pitch (around the *Y*-axis). Roll, pitch, and yaw are the most often used parameters in the navigation literature. One can find the most common expressions of twist, torsion, and rocking in seismology. Tilt is often misunderstood or unclear; one can find several definitions of this term [75]. Basing on [3], twist describes a shear deformation caused by a torsional moment. Torsion means rotation or strain around the vertical axis of the structure. Rocking determines the rotational component around the horizontal axis [3]. Engineers often evolve this term as a rotation of an entire structure around a horizontal axis.

In the seismology literature, the terms “acceleration” and “velocity” implicitly concern translational particle acceleration and velocity. Nowadays, translational acceleration and velocity are specified to differ from rotational velocity. For the same purpose, peak ground rotational velocity, defined as the maximum of the absolute value of the rotational velocity around the three axes, is used in this paper. Nevertheless, sometimes, this parameter is calculated using various formulas. For instance, in [76], the peak ground rotation rate is calculated by $PG\dot{\omega }=max\sqrt{\frac{\omega_{R}^{2}+\omega_{T}^{2}}{2}}$, where *ω_R_* and *ω_T_* denote the radial and transverse rotation rates from 20 s before the origin time of the event to 120 s afterward; on the other hand, in [77], the peak ground rotational velocity at the measuring site is calculated by ${PG}_{RV}{=}_{t}^{max}\sqrt{{RV}_{x}^{2}\left(t\right)+{RV}_{y}^{2}\left(t\right)+{RV}_{z}^{2}(t)}$, where *RV_x_*—horizontal peak ground rotational velocity in the East–West direction, *RV_y_*—horizontal peak ground rotational velocity in the North–South direction, and *RV_z_*—vertical peak ground rotational velocity. These parameters should be always provided with definitions and indications of particular axes. Most of the peak ground rotational velocity values provided in Tables 1–3, 5, 9 and 10 (*PGω_z_*, *PGω_x_*, *PGω_y_*) mean the absolute value of the rotational velocity along the indicated axis to review and compare the data from diversified sources.

### 2.3. Classification of Rotational Motion Regards Sources

The classification of rotation measurements in terms of the maximal amplitudes of the recorded events is extensive, and its results strongly depend on the source of the vibration. One can distinguish the following groups:rotational motions of the ground in the near-source field: Bouchon and Aki [78] recorded a natural fault earthquake at seismological stations located 1–20 km from the fault and 1–50 km from the epicentre. The recorded signal amplitude reached the value of 0.1–1.5 mrad/s. Belvaux et al. [19] presented a rotation with an amplitude of 40–200 μrad/s recorded 6 km from the earthquake epicentre. Takeo et al. recorded a rotation in the near field with an amplitude of 30 μrad/s [30] and 26 μrad/s [79];rotation associated with volcanic eruptions: the signal amplitude of the recorded rotation near the volcano was several dozen μrad/s [80]. Data obtained at the Hawaiian Volcano Observatory by the rotational sensor blueSeis-3A showed the maximal signal amplitude of the order of 2.5 mrad/s during earthquakes associated with large collapse events during the summit eruption [81];rotation recorded during chemical explosions: these events are characterized by a high signal amplitude; the signal was recorded at a distance of 1 km from the explosion and had an amplitude of 38 mrad/s [82];rotation connected with engineering seismology: according to data from [83], rotations with an signal amplitude of mrad and greater are expected;tilt measurements with an signal amplitude of 5 μrad during an earthquake of magnitude M = 6.7 from a distance of 311 km from the epicentre [84];rotation measurements of teleseismic waves were detected using a ring laser gyroscope [34,58,73,85,86,87,88,89,90] with relatively small signal amplitudes of the recorded rotations, from a few nrad/s to 400 nrad/s;measurements of rotation related to the physics of seismological interactions of the order of 10^−8^ rad/s [91]. Studies to identify and separate waves enable better and more modern interpretations of various seismic waves, including the identification of P-waves in opposition to the SV and SH components [92] and the separation of Love and Rayleigh waves;rotation studies in a micromorphic medium with a signal amplitude below 10^−7^ rad/s [93].

The above list, as well as considerations regarding recording conditions, allow the rotation measurement to be divided into the detection of high-amplitude rotation (strong-motion) of the order of tens of μrad/s and more—points 1–4 and rotation with a very low amplitude of the order of tens of 10^−7^ rad/s, or less—points 6–8. The frequency range can reach 10^−4^ Hz to 100 Hz. For near-field observation, they appear to be strong at frequencies around ~0.01–0.1 Hz [34]. In the case of mining tremors, the signals are characterized by a more comprehensive spectral range of vibrations, from 0.5 Hz to over 10 Hz, and it is impossible to isolate one fundamental frequency. This paper performs the division of rotational effect measurements according to its source.

For all the above reasons, the intensification of practical aspects in the field of rotational seismology is challenging due to rigorous technical requirements [94,95]. The sensors for rotational seismology must be characterized by a high sensitivity over a broad frequency range as well as a wide range of detected rotation because the expected ground rotation rate is in the range from 10^−8^ rad/s, even up to a few rad/s, as was shown in the above review. One of the most important attributes of a rotational sensor the is complete insensitivity to linear movements or the ability to measure rotational and translational movements simultaneously and then separate them. The self-noise levels must be temperature-independent, and stability against magnetic field variations is required. The sensors need to be mobile, small, and equipped with an independent power supply to install the sensor in hard-to-reach seismological stations. Generally, two basic methods exist to obtain seismic rotational components: indirectly by numerical conversion of data from dense arrays of classical sensors [54] and directly by carrying out measurements applying appropriate rotational sensors [44].

## 3. Indirect Rotation Research by Numerical Conversion

Table 1 contains the parameters of the rotational signals obtained indirectly by numerical conversion of data from dense arrays of classical sensors presented in the literature from 1982 to 2021. The study by Bouchon and Aki [78] used the discrete wavenumber representation method to analyse the amplitude and shape characteristics of the deformation and rotation wave near the theoretical slip and slip damage embedded in a layered environment. The maximum rotational velocity generated by the 30 km buried strike–slip fault and 1-m slip was calculated to be approximately 1.5 mrad/s, and the peak ground rotation was approximately 0.3 mrad. This paper provided the impetus for Cao and Mavroeidis [96], who developed strategies for the kinematic modelling of potential strike–slip earthquakes by simulating the time histories of ground deformation and rotation near the contact zone using finite differential translational motions generated at very closely spaced stations. The maximum peak ground rotation ranged from 20 to 300 μrad depending on the strike–slip earthquake scenario.

Huang [57] presented calculated rotations from translational velocities by numerically integrating accelerograms from a dense acceleration system at the Li-Yu-Tan Dam, located 6 km north of the Chi-Chi earthquake fault. The amplitude for each of the three axes ranged from 40 to 300 μrad/s.

Stupazzini et al. [97] simulated the rotational wave field induced by an earthquake of magnitude 6.0 and 4.5 in the Grenoble Valley (French Alps) using 3D numerical modelling to replicate the rotational wave field generated by strike–slip earthquakes in the near field. The assumed peak ground rotation for receivers located on soft ground was approximately one mrad, and the maximum ground rotation speed was ten mrad/s.

**Table 1 sensors-24-07003-t001:** Parameters of the rotation (selected maximum value) obtained indirectly by numerical analysis. Legend: Y—year of publication, Ref.—reference, F—frequency, ES—earthquake source mechanism, M_w_—magnitude, R—epicentral distance, PGV_h_—peak value of horizontal ground velocity, PGV_v_—peak value of vertical ground velocity, PGω_z,x,y_—peak value of rotational velocity around the particular axis.

Y	Ref.	F [Hz]	ES	M_w_	R [km]	PGV_h_ [m/s]	PGV_v_ [m/s]	PGω_z_ * [μrad]	PGω_z_ [mrad/s]	PGω_x_ * [μrad]	PGω_x_ [mrad/s]	PGω_y_ * [μrad]	PGω_y_ [mrad/s]
1982	Bouchon and Aki [78]		strike–slip fault	6.6	1	1/1.6	-	200/ 300	1.2/1.5	700/800	-		
2003	Huang [57]	<1.0	The 1999 Chi-Chi, Taiwan earthquake (thrust fault)	7.7	6	0.33	0.50	171	0.385	44	0.126	177	0.331
2008	Spudich and Fletcher [98]	<3.6	2004 Parkfield, California, earthquake and aftershocks (strike–slip fault)	6.0	8.8	0.25	-	88.1	1.09	68.9	0.925	-	-
				4.7	14.0	0.013		4.69	0.0944	4.74	0.0926		
				5.1	14.4	0.060		20	0.446	0.177	0.372		
				4.9	18.3	0.027		13.6	0.247	9.73	0.215		
2009	Stupazzini, et al. [97]	<2	valley of Grenoble, French (strike–slip)	6.0	0.02–0.90	0.4	0.3	1 690	8.24	4000	8.66	1310	0.6
2009	Wang, et al. [99]	<0.5	Newport–Inglewood strike–slip	7.0	<80	-	-	-	0.05–1.5 *		0.05–0.350 *	-	0.05–0.6 *
2019	Cao and Mavroeidis [96]		hypothetical strike–slip earthquake	6; 6.4; 6.8; 7.2; 7.6	1–50	<0.72	<0.24	69.2–194.2		16.9–94.3	-	22.7–98.5	-
			dip-slip earthquake	6; 6.4; 6.8; 7.2; 7.6	1–50	<0.66	<0.93	54.1–144.3		117.9–440.6.9	-	114.2–325.3	-
2021	Cao and Mavroeidis [100]	<1.0	Izmit earthquake 1999	7.5	1–50	0.11–1.26 *	0.03–0.54 *	52.6–471 *	-	6.2–162 *	-	10.7–123 *	-
			2004 Parkfield	6.0	1–50	0.005–0.39 *	0.003–0.18 *	5.6–75 *		2.5–63 *		1.4–48 *	
			1979 Imperial Valley	6.5	1–50	0.06–1.17 *	0.007–0.18 *	21–210 *		9.7–120 *		3.9–48 *	

* PGω_z,x,y_—ground rotation around the particular axis depending on the distance.

Reference [98] provides an estimation of the rotation of the 28 September 2004, mainshock in Parkfield, CA, USA which included an earthquake of magnitude 6.0 and aftershocks of magnitude 4.7–5.1. The data were recorded at 12 accelerograph stations of the U.S. Geological Survey Parkfield (UPSAR) seismic network, consisting of three-component accelerographs located 8.8 km from the San Andreas Fault, USA. Wang et al. [99] simulated several magnitude 7 earthquakes with different sources on the Newport-Inglewood fault embedded in the 3D Los Angeles Basin using a finite-difference method over a frequency range of up to 0.5 Hz. The analysis showed that the variability of the hypocentre leads to significant changes in the ground rotation speed.

Simulations of three well-documented seismic events were analysed in [100]. These include the 2004 magnitude 6.0 Parkfield earthquake, the 1979 magnitude 6.5 Imperial Valley earthquake, and the 1999 magnitude 7.5 Izmit earthquake, analysed by finite-difference simulation translational movements at very closely spaced stations using a kinematic modelling approach.

Based on six-component seismic data, the author of [54] compared three different conversion methods: the traveling wave, the frequency domain, and the difference one. The paper aimed to analyse the characteristics and feasibility of these methods in estimating rotational components using translational observations. The traveling wave and the frequency-domain methods can convert translational components into rotational components, but the frequency-domain method shows greater accuracy. However, the difference method, although it requires denser reference stations, greatly impacts the accuracy of rotational component calculations. The independence of the six seismic components requires rough estimates at more minor deformations, which do not replace accurate observation of the rotational components [54].

## 4. Rotation Effects Recordings During Natural Earthquakes

Natural earthquakes have their source mainly in processes occurring on faults, i.e., large dislocations in the Earth’s crust. This is where the rock blocks move, causing characteristic vibrations felt by people as seismic tremors. There are two main mechanisms for earthquakes on faults: an increase in stress within a rock mass that exceeds the friction on the fault surface between adjacent rock blocks or a reduction in friction on the fault surface. The latter scenario is observed where water enters the fault naturally or due to human activity, leading to increased rock mobility on the fault and potentially triggering an earthquake. One records earthquakes not only on faults but also when magma in the volcano chamber moves and pushes rocks apart. Human activity provides even more such examples, such as when one injects water under pressure into a rock to extract, for example, gas from it. In this way, one changes the stress in the rock mass and provokes tremors.

One of the first successful direct recordings of rotational motions was carried out by Robert Nigbor [82]. In 1994, Robert Nigbor used the Systron Donner, Concord, USA, triaxial gyro sensor to record rotation generated by a powerful explosion of 1 kT explosives [82]. It will also be presented in the section “Recordings associated with artificial explosions”, see Table 5. Nevertheless, the author would like to mention the sensor used. The three commercial rotational micro-electro-mechanical-system (MEMS) sensors represent modern mechanical angular sensors based on highly miniaturized microelectromechanical devices. MEMS gyroscopes apply the Coriolis force to detect the angular velocity, which is related to the acceleration that the body must experience to stay on the rotating surface. Several existing technological solutions exist for these device’s construction, e.g., vibrating tuning fork gyro, vibrating-wheel gyro, resonant wheel gyro, hemispherical resonant gyro, and Foucault pendulum gyro. Applying a similar recording system, Takeo [59,101] (see Table 2) measured signals in the near-source region of the earthquake swarm at the offshore area of Ito in the Izu Peninsula, Japan. He showed that recorded rotational motions are several times larger than those simulated by Bouchon and Aki [78] (see Table 1) based on the dislocation theory. The maximum recorded rotational velocities around the vertical axis were equal to 3.3 mrad/s (earthquakes with a magnitude of 5.7) and about 8.1 mrad/s for earthquakes larger than 3.5. The observational system has been installed at Cape Kawana, 3.3 km from both Earthquake epicenters. The applied Systron Donner MotionPak triaxial gyro sensor has a flat frequency response to rotational velocities around three axes perpendicularly intersecting each other from DC to 75 Hz, and full-scale output equals ±0.873 rad/s.

Applying the same system, in [102], Takeo presented 200 records in near-field regions with hypocentral distances less than 8 km during an earthquake swarm in April 1998 offshore Ito, Izu Peninsula, Japan. The recorded rotational rate varied from 4 μrad/s to 8 mrad/s depending on the magnitudes (1.2–5) and epicentral distances (1.5–10 km); the maximum amplitude signal in the function of the epicentral distance and earthquake magnitude is presented in Figure 3. The sensitivity of the used sensors in the above literature positions was limited to strong-motion signals obtained in near field and artificial sources.

The R-1 rotational sensor is one of the electrochemical-type sensors. They use liquid as an inertial mass in their construction. Fluid movement is recorded via multilayer platinum electrodes staggered between microporous insulating spacers. The electrolyte can move freely through a flexible diaphragm at each end of the transducer channel. The DC voltage applied to the electrodes creates an ion concentration gradient. Due to the conductivity of the electrolyte, the bias voltage and the associated current generate a concentration gradient only between the electrodes. When the sensor experiences acceleration due to ground movement, liquid flows to the electrodes. It causes a change in the current intensity proportional to the liquid’s flow velocity and the ions’ concentration. This technique is known in the literature as Molecular Electronic Transfer (MET) [93]. Such a transducer is essentially a linear velocity sensor [108] in which a symmetrical arrangement of electrode pairs enhances the linearity of the transducer capable of utilizing a feedback loop. This technology was used to build the angular velocity sensor, which was applied to construct an annular channel filled with electrolytes. When the sensor rotates, the MET transducer placed transversely in the liquid channel is forced to move, which, assuming the inertia of the liquid, is converted into an electrical signal. Exemplary models of such sensors can be found on the manufacturers’ websites (for example, RSB-20 by PMD Scientific Inc. (Boston, MA, USA) [www.pmdsci.com, accessed on 20 December 2023], R-1 and R-2 by Eentec (Vilnius, Lithuania) [www.eentec.com] accessed on 22 September 2024). The tests indicate that the R-1 sensor has a linear sensitivity of 6 × 10^−5^ rad/s/(m/s^2^) and a cross-axis sensitivity of 2% [109]. The quality of the calibration process has been questioned, especially in the low-frequency range (below 1 Hz) [110]. Because the frequency response is not flat above 1 Hz, the dynamic range is only 80 dB as opposed to the declared value above 110 dB [109]. In [94], the thermal stability of the R-1 and R-2 sensors was verified. The obtained deviation of the nominal values of the constant signal in the temperature range of 20 °C–50 °C was 27% for R-1 and 18% for R-2. This allows us to conclude that sensors based on the use of liquids require further technical improvements and solutions. Despite the above, the R-1 sensor was one of the most often used to register rotational motion for local earthquakes [64,105], artificial explosions [110], as well as mining activity seismic areas [5,77]. The peak rotational velocity recorded by the R-1 for 52 local earthquakes (0.004–0.634 mrad/s) at the HGSD station in Eastern Taiwan in the function of the sensor’s distance (14.3–260.4 km) and earthquake magnitude (2.57–6.63) is presented in Figure 4 based on [64].

Rotational sensors designed by the company Applied Technology Associates (ATA), Albuquerque, USA [76] represent a different technology utilizing fluid. The fundamental physical principle of their working design is called magneto-hydrodynamics (MHD). The main part of the sensor is a rotating mass consisting of an electrically conductive fluid and a permanent magnet attached to the device’s housing. If the device rotates, the magnetic flux moves through the conductive fluid with a relative velocity, which creates an electric field between the magnetic flux and the line with the fluid. This interaction, called the MHD effect, causes a voltage difference between the electrode surfaces, which a transformer or other active electronic configuration can amplify. The output voltage is proportional to the angular velocity. In 2017, a proto-seismic magnetohydrodynamic (SMHD) three-component rotational rate sensor by ATA recorded rotational motions of 155 earthquakes of magnitude above 2.0 at a temporary after-shock station in Waynoka, Oklahoma, within 220 km of the station [76]. The experiment lasted about two months, and the highest rotational components were recorded during the earthquake with a magnitude of 4.2 at a distance of 0.5 km from the station. The maximum peak ground rotational rate was equal to 2.11 mrad/s, 1.86 mrad, and 1.12 mrad/s for horizontal and vertical components, respectively. Moreover, the data from rotational sensors have been widely compared with data recorded by a translational broadband seismometer.

In reference [106], an example of rotational events recorded at the Książ observatory in Poland on 6 January 2012, at approximately 200 km from the source is presented. Data were collected by mechanical rotational seismometers and an optical instrument using the Sagnac interferometer during an earthquake with a magnitude of 3.8 near Jarocin, Poland. The summarised data presented in Figure 3 and Figure 4 show that rotational peak amplitudes have an exponential relation to the event magnitude for short (2–10 km) and long (20–260 km) epicentral distances. However, the above statement cannot be confirmed by data presented by Brokešová and Málek [103], who had the opportunity to record rotation rates during the swarm activity in the period from 6 October 2008 to 10 December 2008, in the region of Nový Kostel, Czech Republic, about 15 km north of the town Cheb (Eger). The recorded maximum amplitude of rotation rate was about 0.15 mrad/s, which was generated by the earthquake of local magnitude 2.2 with epicentral distance to the recording station of the order of 4.4 km, and the depth of the earthquake source was about 8.6 km. The results obtained by the 3DOF records were in good agreement with the waveform of the transverse acceleration after proper filtering. According to research, swarms in this region are caused by solutions that penetrate from the upper mantle and the characteristic geological structure. Fluids fill the cracks, rising toward the Earth’s surface until they encounter a barrier in the form of an impermeable layer of granite. Over the years, the solutions accumulate in the fractures and their pressure increases until the stress is released in a series of small earthquakes. In 2008, 2500 shocks with a magnitude up to 3.8 were recorded in a month-long swarm. The 3DOF and 6DOF (Prototype I and II, described in Section 6) sensors have been successfully recording several rotational effects with totally different seismotectonic characteristics in the period 2008–2013 in various regions: Czech Republic (West Bohemia/Vogtland, the vicinity of Prague, the Hronov-Poříčí fault zone), and the Provadia region in Bulgaria, the Gulf of Corinth, Greece, and the volcanic complex of Eyafjalla and Katla in South Iceland. The range of rotation rate peak values recorded in the abovementioned period generated by weak earthquakes was in the range of 0.3–150 µrad/s around the vertical axis recorded by 3DOF sensors and 0.06–400 µrad/s recorded by 6DOF sensors. The horizontal component was recorded by 6DOF sensors in the range of 0.1–700 µrad/s. The signal amplitude varies with the earthquake magnitude (0.3–4.7) and epicentral distance 0.67–290 km (Figure 5).

For a detailed description of these devices and their recordings, authors refer to [111]; some of the event’s parameters are presented in Figure 5.

The high potential of using multi-station 6DOF seismic data recordings on an active volcano was presented in [107] where the research was performed by installing the three broadband seismic stations (Nanometrics Trillium Compact 120 VS and RefTek RT130) together with three blueSeis-3A FOGs at Stromboli volcano, Italy. The results confirmed that SV and SH waves greatly contribute to the Stromboli volcano’s wave field, consistent with previous array-based findings. Additionally, by locating the signals and combining gyroscope and seismometer measurements, one can better understand the polarization properties of these waves. The recording of wavefield gradients brings many benefits but also challenges, especially with local changes in velocity and topography. The research shows clear differences between the three groups of volcanic events, one of which may be difficult to detect with traditional seismometers. This discovery highlights the need for further research into using 6DOF techniques in the monitoring of volcanic activity.

## 5. Teleseismic Waves Recordings

Large ring lasers are the most sensitive sensors for ground rotation detection, and they play a significant role in rotational seismology, especially in teleseismic wave recordings. These devices were initially constructed to monitor the Earth’s absolute rotation rate [112], but further, they brought a wide range of local earthquake observations and teleseismic wave research (Table 3). The large ring lasers detect the beat frequency of two counter-propagating laser waves, which is proportional to the rotation rate component perpendicular to the sensor’s active area. Since the early 60s, they have been leaders in inertial navigation and motion control as ring laser gyroscopes (RLGs). They offer a wide dynamic range, high precision, and small size, and they do not require any moving mechanical parts [113]. RLGs used for inertial navigation usually have an area < 0.02 m^2^, corresponding to a perimeter of 30 cm or less. Large ring lasers were built with a much larger perimeter to increase sensitivity beyond navigational RLGs. Several papers have been mentioned to maintain a historical chronology that includes data gathered by this technological solution.

Stedman presented the possibility of measuring local rotational effects of seismic waves by a ring laser in 1995 during a regional earthquake [115]. Professor Hans Bilger of Oklahoma State University, Stillwater, USA designed the applied C-I He-Ne ring laser system. It was built at the University of Canterbury, Christchurch, New Zealand [116,117]. The system included a rectangle of four supermirrors with nominal 99.9985% reflectors and He–Ne laser. It enclosed an area of 0.755 m^2^ and was one of the first ring lasers unlocked at the Earth rate. In 1999, ring lasers recorded rotational components of teleseismic surfaces and body waves from two strong events with magnitudes 7.0 and 7.3 [88]. C-II and G0 were installed in a cavern 30 m underground at Cashmere, Christchurch, New Zealand [88]. The system named C-II has been mounted locally horizontally, and the rotation rate of the local vertical axis has been measured. Hence, it was sensitive to SH or Love wave rotation. During the experiment, the G0 detected a rotation rate about a local horizontal axis. Hence, it detected SV or Rayleigh wave rotation. The maximum signal amplitude of the observed rotation was of the order of 10 nrad/s (Figure 6).

C-II was a slightly larger device than C-I with similar construction, but its improved mechanical monolithic construction and mirror design assured high stability of less than 1 part in 10^7^ over weeks and months. The G0 laser was 3.5 m on its side, and it was a kind of construction bridge between C-II and the G-ring, which was installed at Wettzell in Bavaria, Germany, in 2002. G0 was only expected to demonstrate a single longitudinal mode operation on a cavity much larger than C-II. However, G0 performed remarkably well as the Sagnac gyroscope, achieving a sensitivity of 0.0116 nrad/s/√Hz, and because much larger structures such as UG-1 (366.8 m^2^) and UG-2 (834 m^2^) became constructed in Christchurch, New Zealand. UG-2 obtained a sensitivity of 0.0078 nrad/s/√Hz. However, the sheer physical dimensions of these lasers raised many problems, e.g., out-gassing hydrogen due to stainless steel pipes used for construction, astigmatism, and aberration of beams. Nevertheless, in [118], one can find data indicating that C-II, UG1, and G-ring have all observed daily motions of the Earth’s rotation axis at the poles driven by the effects of lunar gravity on the inhomogeneous mass distribution of the Earth. The G-ring is a monolithic square ring laser with a perimeter of 16 m constructed on top of a large disk of Zerodur. Its area is 16 times larger than in the C-II; therefore, it was much more sensitive to rotation (0.012 nrad/s√Hz). In 2005 [58], Igel et al. presented recordings of weak rotational motion excited by the 2003 Tokachi-oki, Japan, earthquake at the far-field station using 30 rings with a maximal absolute rotation rate of approximately 35 nrad/s. The recordings of rotation around the vertical axis gathered by the G-ring presented in [34] for several earthquakes showed a peculiar consistency between rotational ground motions around the vertical axis and transverse acceleration. Moreover, the G-ring provided certainty that this technology is suitable for seismological observations. It provided a stimulus to construct the ring laser system named the GEOsensor, which was explicitly designed for seismology with a similar sensitivity [119]. After experimental laboratory tests, the GEOsensor was installed in Pinon Observatory, CA, USA. The size of the ring laser was not limited by the design, and it could be customized according to the available space at the host observatory. GEOsensor has recorded data gathered during several earthquakes presented in [114], with a length of 1.6 m on its side, which provides a total area of 2.56 m^2^. Another identical design to the GEOsensor, named PR-1, was located in Christchurch, New Zealand, and mounted vertically on the seventh floor of a high-rise building (the Rutherford building on the central campus of the University of Canterbury, New Zealand). Its role was to monitor the dynamics of the building influenced by external perturbations. It recorded S- and P-waves during an earthquake on 1 February 2008, which was an estimated 1580 km distance from the earthquake source [120]. The data in [120] shows that the tilting effect is significant during the earthquake.

G-Pisa, GINGERino, and GP-2 were prototypes heading for the construction of GINGER (Gyroscopes IN General Relativity) as a result of a collaboration of the University of Pisa, Italy, Technical University of Munich, Germany, and University of Canterbury, New Zealand for the terrestrial detection of the Lense–Thirring effect using ring laser gyroscopes [121]. G-Pisa had a heterolithic design based on the GEOsensor with dimensions of 0.9 m × 0.9 m to 1.4 m × 1.4 m [121]. GINGERino has a side length of 3.6 m and is mounted on a granite support. GINGERino is located in Laboratori Nazionali del Gran Sasso, Italy (LNGS) to determine the noise and stability of the underground laboratory. GP-2 functioned as crucial instrumentation for preventing fluctuations in the laser cavity dimensions [122]. GINGERino has been recording earthquake-generated rotational motions from October to November 2016 associated with an energetic seismic sequence at a local distance. The rotational sensor has been installed with a broadband seismometer inside the Laboratori Nazionali del Gran Sasso, Italy, the underground laboratory of the Italian National Institute for Nuclear Physics. The analysis of dozens of recorded events showed that peak values of rotation rates and horizontal acceleration are markedly correlated. This suggests that rotation might scale with distance, magnitude, site geology, and fault type, like the scaling of peak velocity and peak acceleration in empirical ground–motion prediction relationships. Peak values of rotation rate around the vertical axis for 33 presented events ranged from 6.14 × 10^−7^ to 1.74 × 10^−5^ rad/s [90] (Figure 7). As shown in Figure 7b, the rotational peak amplitudes have an exponential relation to event magnitude, like the results discussed in the previous section.

The most significant interest should be directed at the system ROMY (ROtational Motions in seismologY), which was constructed in 2016 at the site of the Geophysical Observatory in Fürstenfeldbruck, Germany. It is unprecedented in its scale construction, including a four-component large-scale ring laser array [86,123]. Each component is performed as an equilateral, triangular ring laser and arranged to form a downward-pointing tetrahedron. Three triangular gyroscopes have a length of 12 m on each side, while the sides of the horizontal ring on the top are shorter by 1.0 m to fit rigidly on the concrete monument. Figure 8 shows the top view from the ground and the system’s schema [124].

The resolution of the measured signal of a ring laser gyroscope is proportional to the ratio of the quotient of area and perimeter enclosed by the beam path. The main parameters of the above-mentioned large ring lasers are included in Table 4.

The rotational component of teleseismic surface wave observation has also been carried out in the Laser Interferometer Gravitational-Wave Observatory (LIGO) Hanford Observatory (LHO), USA. Data have been recorded by a rotation sensor, denoted in [89] as a beam rotation sensor (BRS), and an array of STS-2 seismometers from Rayleigh waves of six teleseismic events from different locations and with magnitudes ranging from 6.7 to 7.9. BRS includes a meter-scale beam balance suspended by a pair of flexures with a resonance frequency of 10.8 mHz. The angle concerning the sensor’s platform is measured using an autocollimator. This sensor can resolve ground rotation angles of less than one nrad/√Hz above 30 mHz and 0.2 nrad/√Hz above 100 mHz around the single horizontal axis [140]. In the frequency band ranging from 10 to 100 mHz, the BRS has comparable sensitivity to the angle sensitivity of C-II. The analysed data show the possibility of resolving local seismological parameters by rotation and translation components recording from a single station.

## 6. Recordings Associated with Artificial Explosions

Another type of data source is rotation effects recorded during artificial explosions (see summarised data in Table 5). It was commenced by Robert Nigbor in 1994 by recording a rotation generated by a powerful explosion of 1 kT explosives during a non-proliferation experiment at the Department of Energy, Nevada Test Site, USA [82]. He developed the first prototype of a 6DoF strong-motion accelerograph system. Three single-axis analogue gyroscope (Systron Donner, Concord, USA, Model QRS11-00010-200) has been used in [82], which had frequency response between DC and 60 Hz. The recorded peak ground rotation rates around the vertical axis reached 38 mrad/s at a distance of 1 km generated by the underground explosion.

Wassermann et al. [110] used a seven-element seismic array with an R-1 rotational sensor at the array centre to record rotation generated by the demolition blast of a 50 m high building in Munich, Germany, at a distance of about 250 m. The blast consisted of 150 kg of explosives fired sequentially to reduce ground tremors. The observed seismic wave occurred at a frequency range of 1–8 Hz. The comparison of the array-derived measurements with the measured signals showed reasonable results for at least the higher frequency portions of the analysed signals. The authors underlined doubts about the quality of the R-1’s noise level, especially in the lower frequency (<1 Hz). The peak rotation velocity reached about 0.5 mrad/s around the horizontal Y-axis. The peak rotation velocity of about two orders of magnitude higher than that observed in [110] has been recorded during the TAIGER (TAiwan Integrated GEodynamics ReSearch) experiment [141] and by the R-1. There were two explosions set off with 3000 kg and 750 kg explosives. The distance between the explosions and the place of instrument installation varied from 250 m to 600 m. The highest peak ground rotational velocity has been recorded around the X-component at a distance equal to 254 m: 2.74 mrad/s and 1.75 mrad/s for 3000 kg and 750 kg explosives, respectively. The values of the recorded peak ground rotation and peak ground translational acceleration were only about one and a half times larger for the first explosion, even though the first shot used explosives four times larger than the second one.

METER-03 presented in Figure 9, was used to record data in [142]. According to manufacturer specifications, it has a noise floor of 5.7 × 10^−7^ rad/s and a flat frequency response of 0.05–50 Hz. Data have been compared with data derived from records of magnetometers and geophones. Magnetometers work according to Faraday’s law. Copper wire is wound around a magnetically permeable core. When there is a change in the magnetic flux perpendicular to the cross-section of the coil, a current is induced in the wires. The ignition of the Betsy gun was the source of seismic events in Silver Lake, CA, USA.

**Table 5 sensors-24-07003-t005:** Parameters of the recordings associated with artificial explosions. Legend: Y—year of publications, VS—vibration source mechanism, R—distance between sensor installation and source of vibration, PGV_h_—peak value of horizontal ground velocity, PGV_v_—peak value of vertical ground velocity, PGω_z,x,y_—peak value of rotational velocity around the particular.

Y	Ref.	VS	Sensor	R [km]	PGω_z_ [mrad/s]	PGω_x_ [mrad/s]	PGω_y_ [mrad/s]
1994	Nigbor [82]	1 kT chemical explosion at the Nevada Test Site	QRS11 (Systron Donner)	1	24	38	-
2009	Wasserman et al. [110]	Demolition blast of building in Munich, Germany	R-1, eentec	0.2	0.02	0.008	0.05
2009	Lin et al. [141]	3000 kg explosives, TAIGER experiment, Tawian	R-1, eentec	0.2539–0.6082	0.268–0.966	0.370–2.741	0.627–2.524
		750 kg explosives, TAIGER experiment, Tawian			0.301–0.563	0.235–1.750	0.394–1.185
2013	Brokešová and Málek [104]	medium-size quarry blast, 3044 kg explosive, Czech Republic	6 DOF Rotaphone	0.362	~1	~4.5	~2
2018	Barak et al. [142]	Ignition of Betsy gun at Silver Lake, California	METR-03	<1	-	<0.1	<0.2
2019	Kurzych et al. [75] Teisseyre et al. [75,143]	Digging shafts with the multiple blasts technique, Książ, Poland	FOSREM, TAPS, RS.LQ–RP/P	0.075	0.05–1	-	-
2021	Bernauer et al. [32] Kurzych et al. [144,145]	500 g explosive, Fürstenfeldbruck, Germany	BlueSeis-3A, FOSREM, ROMY, Rotaphone-CY, FARO, PHINS, Quadrans, MEMS gyroscopes (Horizon, Gladiator)	~0.05	~0.5 (BlueSeis-3A) ~1 (FOS5-01) ~0.5 (FOS5-02) <0.5 * (BlueSeis-3A) ~0.005 * (ROMY) <0.02 * (FARO) <0.025 * (FOS5) ~0.025 * (PHINS) <0.025 * (Quadrans) <0.05 * (Rotaphone)	<0.1 * (BlueSeis-3A) <0.15 * (PHINS) < 0.1 * (Quadrans) <0.09 * (Rotaphone)	~0.1–0.15 (BlueSeis-3A) <0.15 (PHINS) <0.15 (Rotaphone) <0.15 * (BlueSeis-3A) ~0.15 * (PHINS) <0.15 * (Quadrans) <0.15 * (Rotaphone)
		VibroSeis truck, Fürstenfeldbruck, Germany	FOS5-1	0.096	0.0177	-	-
				0.105	0.0252		
				0.113	0.0386		
				0.121	0.0158		
				0.130	0.0156		
				0.138	0.0141		
2021	Cao et al. [146]	near field explosion, China	RotSensor3C	0.150	~11	~11	~16
2022	Brokešová and Málek [147]	medium-size blast at the Klecany quarry, Czech Republic	Rotaphone, R-1, ADR (array-derived-rotation)	0.240	~0.05 (Rotaphone) ~0.01 (R-1) ~0.05 (ADR)	~0.25(Rotaphone) ~0.1 (R-1) ~0.25 (ADR)	~0.15 (Rotaphone) ~0.03 (R-1) ~0.1 (ADR)
					~0.05 (Rotaphone) ~0.03 (R-1) ~0.06 (ADR)	~0.25 (Rotaphone) ~0.2 (R-1) ~0.22 (ADR)	~0.2 (Rotaphone) ~0.08 (R-1) ~0.1 (ADR)

* 30 Hz low pass filtered waveforms.

The prepared line where the shots took place was 100 m with 5 m spacing between shots. It caused a P-wave with a velocity of 1420 m/s and two surface waves propagating at velocities 135 m/s and 250 m/s recorded by a geophone distance of about 1 km. The recorded signal around the *Y*-component (max. 0.2 mrad/s) of rotation was higher than the maximum amplitude of the signal around the *X*-component, which the authors expected according to the rotational deformation caused by the Rayleigh wave in the analysed survey (poor rotation around the inline axis).

A team from the Military University of Technology, Poland, also studied the artificial rotational events. The authors’ team has been working on the rotational sensor since 1998. The fibre-optic system for rotational events and phenomena monitoring (FOSREM) belongs to the optical group of rotational sensors. It applies the technical realization of the interferometer based on the Sagnac effect. It is performed according to a minimum open-loop fibre-optic gyroscope configuration, where the Sagnac effect produces a phase shift between two counter-propagating light beams proportional to the measured rotation [149]. The main advantage of this approach is its insensitivity to linear motions and the direct measurement of the rotation rate. The deeper description of FOSREM’s configuration and software can be found in [75,150,151,152]. The theoretical sensitivity of FOSREM is at the level of 20 nrad/s/√Hz. The experimentally obtained Angle Random Walk (ARW) is equal to 32 and 49 nrad/s for FOSREM-1 and FOSREM-2, respectively [75]. The thermal stability of FOSREM has been experimentally verified at the temperature range from 0 °C to 50 °C during the cooling and heating process, and it is presented in [150] with only a 0.06%/°C output signal instability. The basic parameters of the constructed rotational systems at the Military University of Technology, Poland are presented in Table 6. The main limitation of the above solutions is connected with applying the standard single-mode optical fibre—SMF-28, Corning, Arizona, USA. Such an approach limited the general cost of devices using even 15,000 m optical fibre in one sensor coil. However, SMF-28 is sensitive to temperature and pressure fluctuation, which generates drift in the output signal. For the above reason, all FOSREMs should be calibrated after their installation. The calibration is made remotely via the internet; the recorded seismogram has automatically removed the constant component of the signal by suitable software [153,154].

Two FOSREMs have been mounted together with TAPSs (Twin Antiparallel Pendulum Seismometers) in the geophysical observatory of the Polish Academy of Science in Książ, Poland, which is in the area of mining activity. TAPS was the first of the mechanical rotational seismometers constructed in Poland. It was designed by the modification of horizontal electromechanical SM-3 seismometers [155]. It includes two SM-3 seismometers located anti-parallel on a common vertical axis [155]. It allows for the measurement of the horizontal component of the linear velocity of the axis of the seismometer system, perpendicular to the lever of the seismometers, and on its basis, determines the rotation velocity relative to this axis. It is a Polish example of a rotational sensor that indirectly determines the angular velocity similar to, e.g., a Rotaphone consisting of commercially available geophones installed horizontally or vertically on a joint rigid base. In the spring and early summer of 2018, two FOSREMs and TAPSs recorded rotational events induced by digging two vertical shafts leading underground using numerous multiple bursts. The microshocks were generated by real dynamite explosions aimed at creating tourist tunnels leading to Książ castle, Poland. It should be underlined that these works generated robust oscillations with the peak velocities of the ground motions greater than usually encountered (mainly of copper and coal mining provenance). The distance between the centre of the shafts and rotational seismometer locations was around 75 m.

**Table 6 sensors-24-07003-t006:** Historical brief of the fibre-optic seismograph constructed at the Military University of Technology, Poland; Legend: Y—year of construction, R—radius of the sensor loop, L—length of the optical fibre, T—type of optical fibre, S—sensitivity, Ω_max_—maximal detectable rotation velocity, F—frequency.

Sensor	Y	R [m]	L [m]	T	S [rad/s]	Ω_max_ [rad/s]	F [Hz]	FOG Conf.
GS-13P	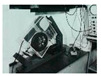 [156]	1998	0.1	380	Hi-Bi	3.49 × 10^−3^	1.74	DC–100	open-loop
FORS-I	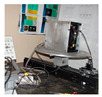 [153]	2001	0.1	400	Panda	2.2 × 10^−6^	4.6 × 10^−4^	DC–100
FORS-II (FOS1), AFORS (FOS2)	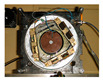 [157] 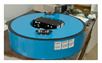 [106]	2004–2010	0.34	11,000 (FORS-II)15,000(AFORS)	SMF	4.2 × 10^−8^ (FORS-II)4 × 10^−9^ (AFORS)	4.8 × 10^−4^(FORS-II)6.4 × 10^−3^ (AFORS)	0.83–106.15
FOSREM (FOS3, FOS4)	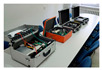 [75,106]	2015	0.125	5000	SMF	2 × 10^−8^	up few rad/s	DC–328.12
FOS5	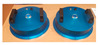 [158]	2018	0.125	5000	SMF	7 × 10^−8^	10	DC–1000	closed-loop
FOS6	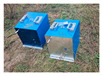 [159]	2023	0.125	6000	SMF	50 × 10^−9^	10	DC–1000

Maximum velocities of the vertical rotation component observed during 16 multiple blasts were much higher than those recorded during mining activity and were in the range of 0.08–35 and 0.01–0.45 mrad/s recorded by FOSREM and TAPS, respectively [143]. Despite the different kinds of rotational sensors, the data from TAPS and FOSREM agreed, which is broadly presented in [75,143]. Figure 10 presents examples of recorded blast-induced angular motions by FOSREM-01 and -02.

Another comparative active experiment has been presented in [147], with data obtained from three methods. The Rotaphone sensor system, the commercial R-1 rotational sensor by Eentec, and a small-aperture array of twelve standard short-period LE-3Dlite velocigraphs by Lennartz Electronic Ltd. (Tubingen, Germany) in a rectangular arrangement have been recording rotation rates generated by a medium-sized quarry blast near Prague, Czech Republic. The three methods used resulted in rotational records that matched only partially; some were only approximately like others. The rotational component from a medium-sized quarry blast at a distance of approximately 240 m reached an amplitude of the order of 10^−5^ and 10^−4^ rad/s for the vertical and horizontal axes, respectively. Generally, the rotation component around the *Y*-axis and *X*-axis were approximately two times and three times, respectively, stronger in amplitude compared to the rotation rate around the Z-axis.

A broader and worldwide experiment is presented in [32]. “Rotation and strain in Seismology: A comparative Sensor Test” gathered more than 40 sensors in the Geophysical Observatory Fürstenfeldbruck, Germany, from 18–22 November 2019. The number and diversity of the rotational sensors made this scientific event the first of its kind. The blueSeis-3A, ROMY, three permanent broadband stations, 80 Channels Geophone system, four Rotaphones, two Gladiator, three Horizon, four Quadrans, one Octans and several accelerometers, giant FOG, giant FOG FARO, Distributed Acoustic Sensing cable, as well as FOSREMs (two type FOS3 and two type FOS5), were mounted in the bunker and in the field in order to record self-noise vibrations generated by artificial explosions within distances range of 50 m to 1.1 km from the instrument installations, as well as vibration generated by a special VibroSeis truck (peak force: 275 kN) placed within a distance range of 20 m to 1.5 km to the instrument installations (Figure 11b). One of the most important conclusions of these studies, which would seem obvious, is the reservation that the devices must have precise amplitude and phase information across the entire waveform. Manufacturers must pay special attention to this shortcoming due to inaccurate timestamp strategies or improper decimal filtering. The obtained self-noise for all portable sensors indicated that FARO (a one-component prototype based on the principle of an open-loop interferometric fibre-optic gyroscope [160]) is the most sensitive. Nevertheless, considering its portability and size, it is only suitable for laboratory use or permanent installation. The blueSeis-3A and Rotaphone-CY (in a narrow frequency band from 1 Hz to 20 Hz) were characterized by the lowest self-noise, and in comparison to FARO, they are portable. For the data presented in [32] concerning recorded during explosions, the maximum amplitude of rotation recorded by all applied sensors did not exceed 100 µrad/s for horizontal components and 500 µrad/s for vertical components.

One of the phases of the experiment involved registering external excitations generated by the VibroSeis truck. The truck with a specific mass is designed to generate complete sine waves. This truck was active for 15 s, generating waves with a frequency sweep ranging from approximately 7 Hz to 120 Hz. During the experiment, the truck stopped six times every 1–2 min to generate excitations. The distance between successive sweeps was 10 m, and the distance between the FOS5 and the operation of the VibroSeis truck ranged from 96 to 138 m. Analysing Table 5, it can be seen that changing the distance of FOS5 from the VibroSeis truck did not affect the maximum amplitude of the recorded signal [144,145]. Such cooperative experiments maintain that all instruments under the test should be subjected to this identical excitation. Therefore, the place of the installation must be carefully selected and characterized. This kind of international cooperation emphasizes the need for standard sensor comparison and analysis to expand rotational seismology knowledge and test new ways to process data by revealing the characteristics of the wave field and source, as well as the location of the source.

The review [95] shows that a fibre-optic gyroscope-based system is arguably the most successful fibre sensor technology today for rotational seismology. This technological solution is utilized to construct one of the rotational sensors by the company Exial (previously iXblue). BlueSeis-3A has been applied to record, inter alai, a volcano-related earthquake [81] or explosion [32]. Another example of the application of this technology is RotSensor3C, constructed by a team in China [146]. It recorded data during the explosion with a maximum signal amplitude of 0.016 rad/s around the *Y*-axis. The comparison of the main characteristics of RotSensor3C, the BlueSeis3A, and R-2 can be found in [146].

The Rotaphone, mentioned earlier, is widely used in rotational seismology. It was constructed by the Charles University team, Czech Republic, led by Johana Brokešová. It is a system consisting of commercially available geophones installed horizontally or vertically on a joint rigid base. Its design enables the measurement of both linear and rotational vibrations. The various types of Rotaphones and their parameters are presented in Table 7. The first prototype, constructed in 2008 and named 3DOF, consisted of six vertical geophones LF-24 (Sensor Nederland B.V., Voorschoten, Holland) mounted on a massive metal disc 0.25 m in diameter at regular intervals. This device recorded rotational and translational seismic components induced by small seismic events in the swarm area in the West Bohemia/Vogtland region (station Květná) [103].

The next type of Rotaphone, named 6DOF, consisted of eight horizontal and one vertical SM-6 geophones (Sensor Nederland B.V., Voorschoten, Holland) mounted onto a cubic-shaped metal frame and equipped with a modern analogue-to-digital converter. This version of the sensor is characterized by a much lower minimum measurable angular velocity of the order of 2.16 nrad/s [161]. The upgraded version of 6DOF, named ’Prototype II’, possessed three additional vertical geophones.

**Table 7 sensors-24-07003-t007:** Parameters of the mechanical rotational sensors constructed by Institute of Geophysics of Polish Academy of Science, Poland (TAPS), and by the Charles University team, Czech Republic (Rotaphone).

Device	TAPS [155]	3DOF [162]	6DOF [162]	D [163]	CY [163]
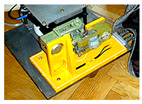	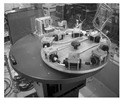	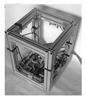	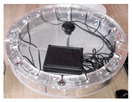	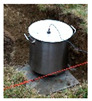
Year of construction	1998	2008	2012	2015	2019
Sensitivity [nrad/s]	100	16.7	2.16	3.77	0.042
Max. detectable rotation [mrad/s]	100	10	287	31.7	31.68
Dynamic range [dB]	120	100	120	120	120
Frequency range [Hz]	0.7–50	1–100	2–60	2–80	1–100
Sampling rate	100	250	250	250	250
Sensors: [quantity × type]Eigen frequency	2 × SM-3 45	8 × LF-24 1	9 (prototype I);12 (prototype II)× SM-64.5	16 × SM-6 4.5	12 × SM-6 4.5
Spacing of paired sensors [m]	0.28	0.3	0.3	0.4	0.3
Operating temperature [°C]	−10–+45	−20–+40	−20–+40	−20–+100	−40–+70
Weight [kg]	15	4.5	9.5	15.3	22
Dimensions [length × width × height] [mm]	450 × 180 × 350	250 * × 10	350 × 350 × 430	445 * × 112	550 * × 500

* disc diameter.

The 3DOF and 6DOF (Prototype I and II) sensors have been successfully recording several rotational effects with totally different seismotectonic characteristics in the period 2008–2013 in various regions: Czech Republic (West Bohemia/Vogtland, the vicinity of Prague, the Hronov-Poříčí fault zone), the Provadia region in Bulgaria, the Gulf of Corinth, Greece, and the volcanic complex of Eyafjalla and Katla in South Iceland. Subsequently, the designers of the 6DOF system used the previous substrate and mounted 16 SM-6 geophones around the disc (eight horizontally and eight vertically), separating each pair of geophones at a distance of 0.4 m. This design was named Rotaphone-D [162]. Thanks to sixteen geophones, the angular velocity components are determined using data from several pairs of devices. This allows for their precise calibration and increases the system’s signal-to-noise ratio so the angular velocity can be defined with higher accuracy. The latest version is named Rotaphone-CY, where a new model of an A/D converter and a more precise placement of the geophones to parallel pairs have been applied. Moreover, better housing has been adapted to protect the device from external electromagnetic noise. Four Rotaphone-CYs participated in the “Rotation and strain in Seismology: A comparative Sensor Test” experiment at Geophysical Observatory Fürstenfeldbruck, Germany [32,163]. There was a series of explosions; the recordings of the Rotaphones have been analysed in [163], and some brief presentations of the record parameters are shown in Table 8.

## 7. Research Concerning Rotational Effects in the Mining Activity Region

In the rotational seismology field, several research studies concerning rotational effects in the mining activity region can be found. The angular velocity of seismic vibrations measured on the surface may significantly influence the impact on buildings with large linear dimensions, e.g., high chimneys and bridges [164,165]. However, there has been no systematic monitoring of seismic rotational vibrations in the near wave field induced by mining exploitation. The reason for this was the lack of mobile intrinsically safe rotation sensors that could be used in underground seismological networks and methane hazard conditions. Rotational effects may significantly impact the stability of underground excavation and the behaviour of roadway supports. Nowadays, more and more examples of rotational events associated with mining activity are presented. Some of these are included in Table 9.

Research on rotational effects associated with mining activity has been widely explored in Poland. Two areas of intensive mining in Poland can be distinguished: Upper Silesian Coal Basin (USCB) [166], and Legnica–Głogów Copper District (LGCD) [167]. These areas served as convenient seismology and earthquake engineering test fields for Zembaty et al. [5], Kurzych et al. [75], Fuławka et al. [77], and Jaroszewicz et al. [158]. Over 1000 seismic phenomena with magnitudes above 1.0 are recorded yearly in LGCD. Some of them are strongly felt on the surface by local residents, resulting in rock bursts and damage to the mine workings. Exploiting coal and copper deposits in the USCB and LGCD leads to mining tremors, causing damage to buildings. The primary mechanism of propagating seismic waves from the source to the surface is the same for earthquakes and mining tremors. This leads some seismologists to conclude that apart from depth and magnitude, there are no fundamental differences between earthquakes and mining tremors [168]. However, from the point of view of surface effects, these are quantitative and qualitative differences. The magnitudes of mining tremors rarely exceed 4 to 5, and earthquakes significantly impact structures located near their epicentres, generally for magnitudes above 5 to 6. A typical earthquake recording lasts 10–30 s (although there are exceptions), while mining tremors typically last a few seconds. The most important difference between mining tremors and earthquakes is differences in the spectral properties of records. Most mining tremors in the LGCD area generate high-frequency and short (1–2 s) acceleration waveforms on the ground surface. According to the latest reports, surface ground rotation can be the result of various phenomena: the significant size of the seismic focus in comparison to the hypocentre distance, hypothetical rotational waves, reflections of body waves from the ground surface, and surface wave propagation.

In the paper by Zembaty et al. [5], one can find a collection of 51 records of ground rotation from a surface measuring station located in the mining area of the Ziemowit coal mine, which is situated in the USCB in Poland. The triaxial rotational sensor R-1 and translational ones by the EA-120 translation sensors have been used. The three strongest events of the whole recorded series have been analysed in detail in [5]. The maximum value of the recorded rotational velocity about the north–south axis equals 0.527 mrad/s, and it corresponds to a maximum acceleration equal to 32.348 mrad/s^2^ for the event with a magnitude of 2.6.

**Table 9 sensors-24-07003-t009:** Parameters of the recordings associated with mining activity; Legend: Y—year of publications, F—frequency range of the sensor used, VS—vibration source mechanism, M_w_—magnitude, R—distance between the event source and sensor, PGV_h_—peak value of horizontal ground velocity, PGV_v_—peak value of vertical ground velocity, PGω_z,x,y_—peak value of rotational velocity around the particular axis.

Y	Ref.	F [Hz]	VS	Sensor	M_w_	R [km]	PGV_h_ [mm/s]	PGV_v_ [mm/s]	PGω_z_ [μrad/s]	PGω_x_ [μrad/s]	PGω_y_ [μrad/s]
2014	Kurzych et al. [169]	0.83–106.15	mining activity, Lubin, Poland, 2011–2013	AFORS	2.3–3.3	70	-	-	6/60	-	-
			earthquake Honshu, Japan, 2011		9	8800			15		
2015	Brokešová and Málek [47]	2–60	geodynamically active region, West Bohemia/Vogtland, 2012, (band-pass filtered 2–24)	Rotaphone 6DOF	2	0.7	0.081	0.02	4	5.7	4
			active rift, Gulf of Corinth, Greece, 2012 (band-pass filtered 1–14)		2.4	6.3	0.326	0.06	10	15	25
			Microearthquake, rifting and volcanic activity in South Iceland, 2014 (band-pass filtered 1–14)		2.3	14.9	0.05	0.025	3.3	1	2.5
2016	Zembaty et al. [5]	0.05–20	mining exploration monitoring, USCB, Poland	R-1	2.6	0.943	20.3	-	491	513	527
					2.5	1.203	8.3		514	425	298
					2.2	0.973	13.8		430	276	500
2019	Kurzych et al. [75]	DC–328.12	seismic shocks induced by the exploitation of copper ore, Książ, Poland, 2017–2018	FOSREM	-	70	-	-	1–20	-	-
2020	Fuławka et al. [77]	0.05–20	tremor in the near-wave field, Rudna-I shaft, Poland, 2019	R-1	-	<7	0.01–4	0.01–4	few μrad/s up to 195 mrad/s *		
			monitoring of the tailing pond, Poland, 2019		-	2.3–8	0.01–4	0.01–4			
2021	Jaroszewicz et al. [158]	DC–1000	mining-induced events, coalmine “Ignacy”, Rybnik, Poland, 2021	FOSERM	-	-	-	-	51.8 (FOS5-01) 60.8 (FOS5-02)	-	-

* depending on the distance from the mining tremor and its energy, peak ground rotational velocity is calculated as *PG*_RV_, as described in Section 2.2.

Kurzych et al. [169], Fuławka et al. [77], and Jaroszewicz et al. [158] are focused on the rotational events recordings in the LGCD. Fuławka et al. [77] applied R-1 and EP-300 seismometers to record data connected with mining-induced seismicity. The authors focused on two stations, which are the most interesting, considering seismic activity: Zelazny Most—one the biggest flotation tailing ponds worldwide and the Rudna-I mining shaft. Sensors were installed at the concrete base near the Rudna-I shaft and in a two-m-deep concrete well located at the dam of the Zelazny Most tailing pond, Poland. High-energy events (E > 10^6^ J) were analysed and located below 8 km from measuring posts. The maximum value of the peak ground rotational velocity, equalling 0.46 mrad/s, in the first station, was generated by a high-energy seismic tremor with energy equal to 3.1·10^8^ J at a distance of 4.446 km from the source. According to the waveforms recorded near Rudna-I, high-energy tremors that occur below 2 km in nearly all cases were related to rotational velocity over one mrad/s. The maximum rotational velocity of the seismic wave reached the value of 195 mrad/s and was caused by a seismic tremor with the energy of 3.6·10^7^ J located at a distance of 1.550 km from the measuring post. According to the distance from the mining shock and its energy, the recorded rotational velocity varies in the range from several μrad/s to 190 mrad/s, which is five times higher than in the case of measurements of the rotation generated by blasting 1 kT of explosive in underground conditions from a distance of 1 km [82]. Studies conducted in various location fields [5] confirmed a strong correlation between the logarithm of the peak ground rotational velocity and peak ground translational acceleration. However, these studies [77] have shown that this relationship is not present at very small distances, less than 2 km hypocentrically and 500 m epicentrically, where the rotational velocity has no relationship to the translational acceleration. A higher dominant frequency and attenuation also characterize the rotational component of the ground motion wave compared to the translational component. Therefore, monitoring and analysing these seismic movements is important, especially in the near field where the expected rotational speed can be high.

Kurzych et al. [169] presented regional seismic mining events of a magnitude range of 2.3–3.3, which occurred in the Lubin area, Poland, with the maximum rotational velocity amplitude reaching 60 μrad/s recorded by AFORS-1. In the period of 12 January 2017–18 January 2018, two FOSREMs recorded two types of signals around the *Z*-axis—torsion and tilt, in the frequency range DC–10.25 Hz [75]. Systems were mounted in the geophysical observatory of the Polish Academy of Science in Książ, Poland. The recorded events were associated with mining activity in this region, where the phenomenon of mining shock is most often associated with rock mass cracking, as well as its collapse or displacement along fault surfaces, making it a source of shock waves. The examples of the recordings are presented in Figure 12.

The tilt was one-directional rotational and caused by a rock mass shock, resulting in an excavation or its part being suddenly destroyed or damaged. The average value of the signal maximum amplitude for sixteen recordings for the tilt phenomenon (61.025 ± 0.097 μrad/s) is definitely higher by approximately one order of magnitude than for the recorded rotational events in the form of torsional motion (61.000 ± 0.009 μrad/s)—see Table 10. This is due to the greater rapidity of the tilt phenomenon, the source of which is most likely the method of exploitation, so-called collapse, generating unexpected violent landslides.

## 8. Engineering Area of Rotational Seismology

Recent research [170,171,172,173] shows that the contribution of the rotational component of vibrations is significant when calculating forces in tall and slender objects, e.g., free-standing chimneys. Such vibrations may be undesirable and harmful to certain building structures, even if these are rotational speeds of the order of milliradians per second. Calculations carried out by Bońkowski et al. [165] using data from the seismological archive of the Upper Silesian Regional Seismological Network of the Central Mining Institute, Poland, showed that for a high chimney, the influence of the rotational component in the entire bending moment response ranged from 18% in the upper part of the object to 65% in the chimney base.

The work [174] analysed the behaviour of the Military University of Technology building, Warsaw, Poland, built in the 1960s. Its structure is a reinforced concrete frame with reinforced concrete beams and ceiling slabs supported by beams. The external walls are filled with a reinforced concrete frame structure using silicate bricks with plaster on both sides. The analysed building was approximately 123 m from a single-carriageway road with railway tracks. The FOSREM sensor was installed on each building floor, and a series of angular velocity measurements were gathered around a vertical axis. The collected data were processed, and the most minor noisy signals from the night were selected. The tests showed that the vibration signal amplitude decreases as the building height increases. The building construction attenuates vibrations on the upper floors (58.4 μrad/s—ground floor; 56.1 μrad/s—1st floor; 58.1 μrad/s—2nd floor) but only up to the third floor (59.4 μrad/s), where the amplitude begins to increase [174]. This is probably because, at the highest level, the structure of the building does not stabilize its operation at this level. The highest maximal angular velocity value was recorded in the basement, equalling 64.5 μrad/s. It is not easy to clearly point out the source of vibration, i.e., mechanical devices, traffic vibrations, construction works, or environmental conditions.

The analysis of the amplitudes of translation accelerations and rotation rates along with their variations over time at the top and bottom of the Grenoble city hall, France, is presented in [175]. Apart from permanent accelerometric translation sensors, the BlueSeis-3A rotation sensor has been installed at the top of the building to analyse the influence of ambient vibrations. Under ambient vibrations, a large ratio is observed at the ground level between acceleration and rotation compared with even moderate earthquake conditions. The presented example of 10 min recordings of rotation under ambient vibrations reaches the absolute maximal value of angular velocity equal to 5 μrad/s, while at the time of the local storm, it is equal to 0.1 mrad/s.

In [176], the authors investigated the application of rotation rate sensors in vibration-based damage detection in a laboratory environment. The cantilever plexiglass beam under kinematic excitations was investigated theoretically and experimentally. The differences in the response recorded by Horizon (HZ1-100–100) between intact and damaged beams were seen. The authors proved that rotation rate sensors can be effectively calibrated to monitor even slight variations of the flexural stiffness of beams by using the sensors located on beams or frames.

The authors of [177] investigate the effects of ground rotation on engineering structures using a numerical model of the Grand Chancellor Hotel in Christchurch, New Zealand, which was severely damaged during an earthquake. The importance of ground rotational motion on the coherent translational motion of buildings was investigated, taking inter-story drift as an indicator of structural damage. The results indicate an increase in inter-story drift by up to 15% when considering the rotational component of ground motion. According to damage observations, rotations appear essential for tall buildings in the earthquake area. The effects of rotational motions on structures should be further investigated to avoid underestimating drifts.

## 9. Discussion

As seismology is an observational science, high-quality data are crucial. Therefore, it is essential to develop two classes of instruments: for observations using teleseismometers and for use in dense seismic networks. In seismology, it is observed that the entire seismic network is a measuring device, not a single sensor, and the measurements aim not at ground movements but at the parameters of the earthquake source. Despite continuous development, rotational seismology now allows for observing the rotational movements of the Earth’s surface, which provides valuable information. In the case of observations at teleseismic distances, the seismic wave field can be approximated by plane waves, which significantly facilitates the analysis. While large ring lasers and seismic arrays are expensive, even single sensors can provide essential data. In the near field, on the other hand, it is necessary to collect large amounts of data to draw reliable conclusions. Here, the amplitudes of rotational movements are higher, allowing networks to be constructed based on inexpensive sensors. Seismic networks should be deployed in earthquake-prone areas to ensure the comprehensive monitoring of potential hazards. It is also crucial to cover the nearest seismic field with observations.

The development of rotational seismology remains an area of intense research, generating more questions than answers. The future of this field will depend on effective data collection, improvement of measurement instruments and methods, and the development of the theory of rotational motions. Despite larger and larger amounts of data and papers, there are still several questions to be solved, i.e., the way to store rotational energy, the fault zone, the relationship between rotational elasticity and normal elasticity, the nature of the dynamic breakdown of granular media, the existence of rotational creep, laws governing the damping of rotational force.

Combining the translational and rotational components in interpreting a seismic signal is vital for many reasons. Firstly, it helps improve the signal-to-noise ratio in seismic observations. Secondly, it improves the quality of linear displacement seismic records by correcting the instrument’s response to rotational motions. Additionally, the joint consideration of the translational and rotational components helps to unambiguously determine the location of the fault plane within the source when examining source properties and media structure and increases the capabilities of seismic tomography. In the context of the analysis of rotational movements in seismic zones, it is necessary to study their distribution and values, as this impacts the assessment of the behaviour of buildings during earthquakes and the study of the influence of soil properties and nonlinear effects. Research on building structures using rotational motion sensors reveals their critical sensitivity to such movements and determines rotational motion modes and resonance frequencies. Rotational data analysis is also used in volcanology. The recorded vertical component of rotational movement provides additional observations of Earth’s free toroidal oscillations that are difficult to detect with standard sensors, and rotation sensors are used to correct the readings of precision instruments sensitive to rotational motions, as well as to search for gravitational waves. In seismic surveys, the use of rotational sensors increases the ability to separate the P- and S-waves and detect the arrival of the surface wave with high accuracy. Finally, rotational seismic sensors have great potential for the remote monitoring of underground drilling equipment, which can be important in mining activities. Taking into consideration all the above-discussed issues, one needs to underline the interdisciplinary character of rotational seismology, which touches both seismological aspects associated with earthquakes as well as structural health monitoring. Moreover, machine learning applications for rotational seismology should be also involved, which can be excellent for earthquake catalogues, parameter analyses, and ground motion predictions.

## Figures and Tables

**Figure 1 sensors-24-07003-f001:**
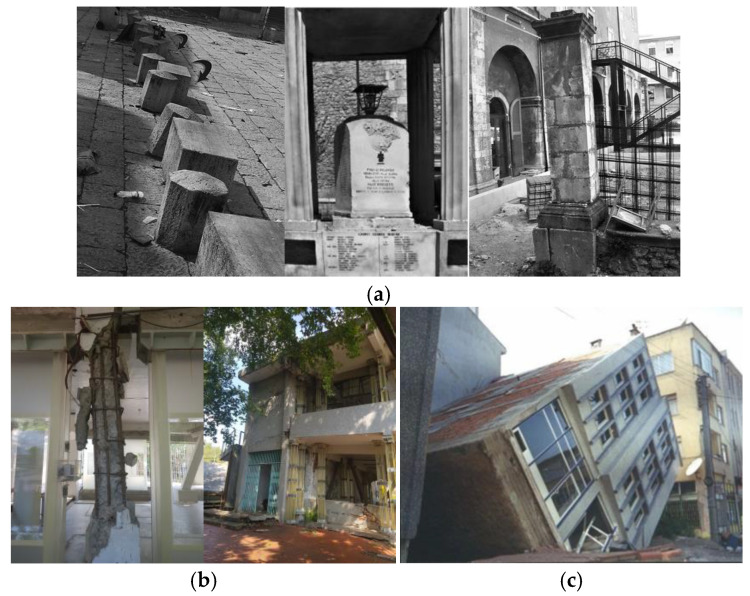
Observation of structural damages after earthquakes: (**a**) pictures of rotated objects in downtown L’Aquila with non-coherent directions of rotation (both clockwise and counter-clockwise) caused by the 2009 L’Aquila (central Italy) earthquake [12]; (**b**) damages in buildings after 21 September 1999, a strong earthquake of 7.3 in the central part of Taiwan, presented in the 921 Earthquake Museum of Taiwan; and (**c**) example of an overall rotation of the base of the structure with an overturning motion [14] during 1999 Kocaeli earthquake, Turkey.

**Figure 2 sensors-24-07003-f002:**
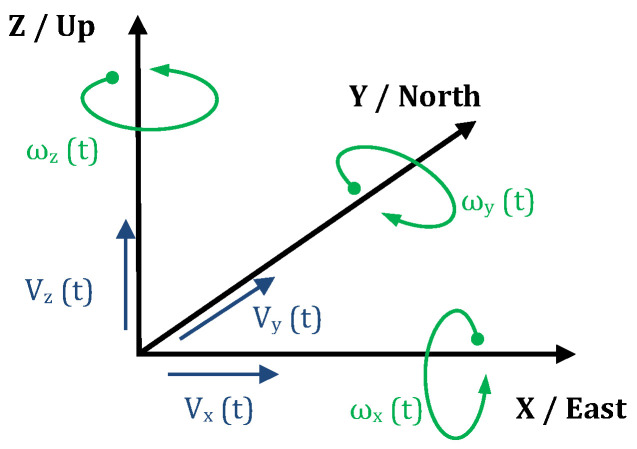
The Cartesian coordinate system for translational (*V_x_*, *V_y_*, *V_z_*) and rotational (*ω_x_*, *ω_y_*, *ω_z_*) velocity components.

**Figure 3 sensors-24-07003-f003:**
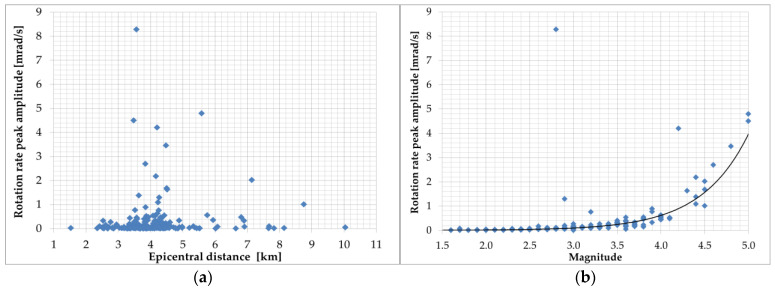
The rotational rate peak amplitude versus epicentral distance (**a**), and magnitude (**b**) based on data presented in [102].

**Figure 4 sensors-24-07003-f004:**
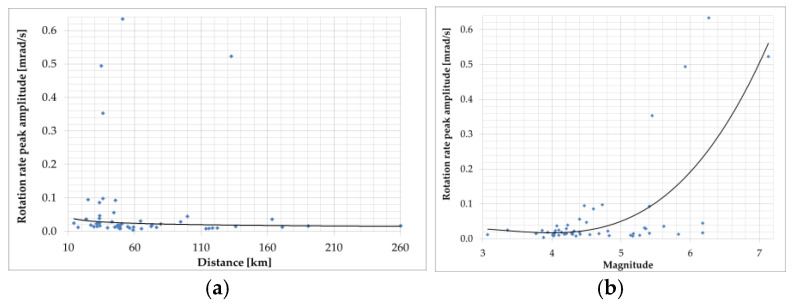
The peak rotational velocity recorded by the R-1 based on [64] for 52 local earthquakes at the HGSD station in eastern Taiwan as a function of (**a**) sensor’s distance and (**b**) earthquake magnitude.

**Figure 5 sensors-24-07003-f005:**
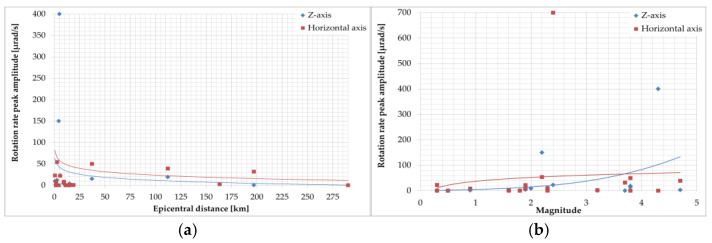
The rotational rate peak amplitude versus epicentral distance (**a**) and magnitude (**b**) based on data presented in [111].

**Figure 6 sensors-24-07003-f006:**
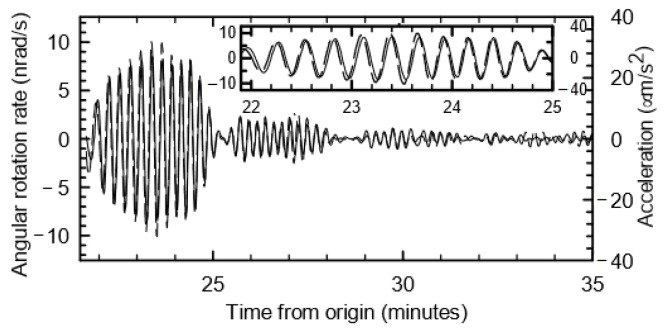
Recording of rotational component around the vertical axis (solid line) and transverse acceleration (dashed), measured by C-II and EARSS/40T, respectively, during the New Ireland earthquake, 19 January 1999 03:35:33.8 (magnitude 7.0) [88].

**Figure 7 sensors-24-07003-f007:**
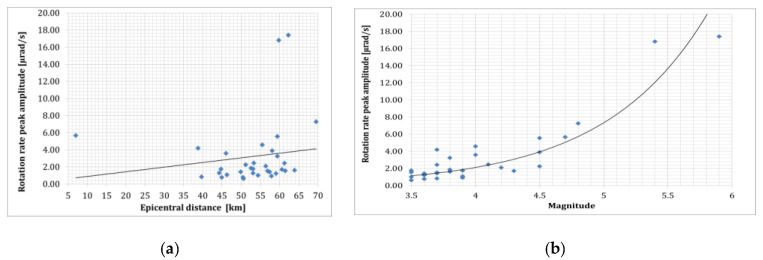
The rotational rate peak amplitude versus epicentral distance (**a**) and magnitude (**b**) based on data presented in [90].

**Figure 8 sensors-24-07003-f008:**
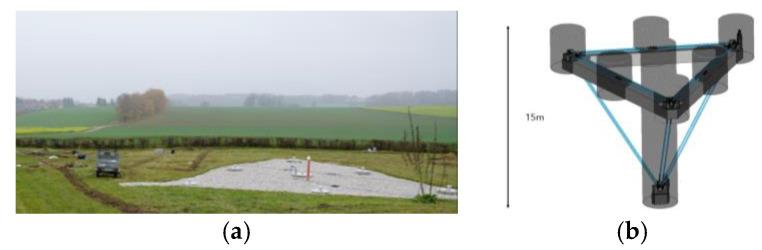
ROMY at the Geophysical Observatory, Fürstenfeldbruck, Germany: (**a**) top view from the ground, (**b**) the schema of the ROMY [124].

**Figure 9 sensors-24-07003-f009:**
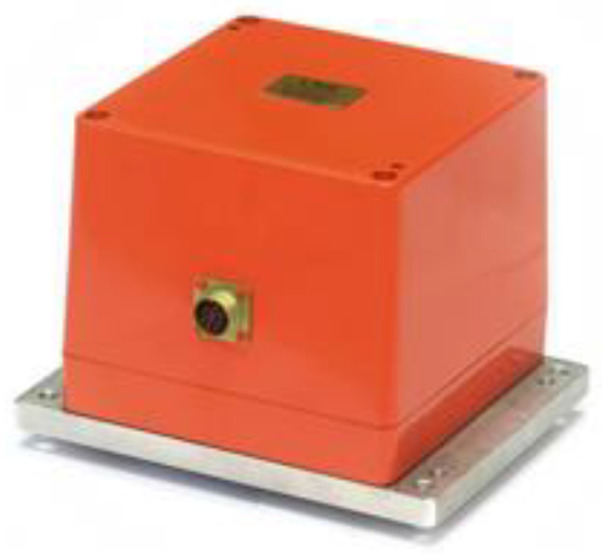
Rotational seismometer METR-03 [148] used in the experiment presented in [142].

**Figure 10 sensors-24-07003-f010:**
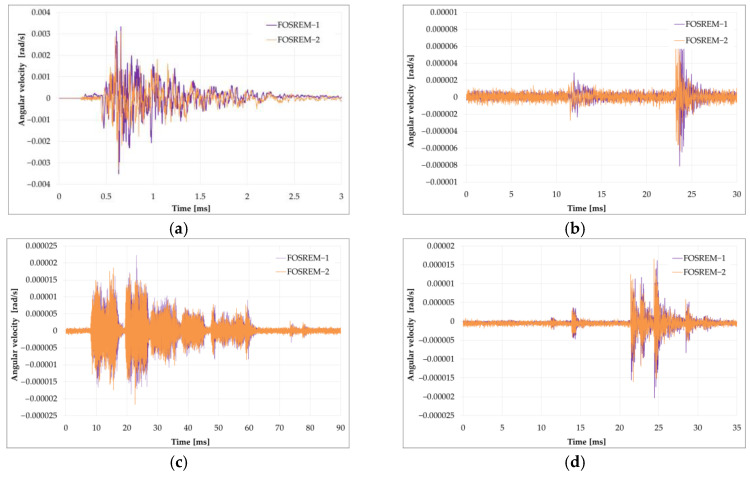
Examples of data recorded during microshocks by FOSREM-1 and -2 in Książ, Poland with an absolute maximum signal amplitude equal to: (**a**) 3.41 mrad/s (FOSREM-1), 3.13 mrad/s (FOSREM-2); (**b**) 0.0081 mrad/s (FOSREM-1), 0.0075 mrad/s (FOSREM-2); (**c**) 0.022 marad/s (FOSREM-1), 0.021 mrad/s (FOSREM-2); (**d**) 0.02 mrad/s (FOSREM-1), 0.017 mrad/s (FOSREM-2).

**Figure 11 sensors-24-07003-f011:**
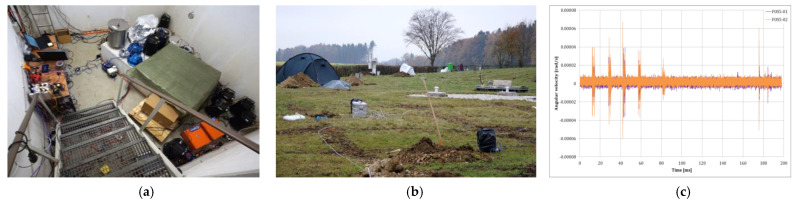
“Rotation and strain in Seismology: A comparative Sensor Test“, which took place in Geophysical Observatory Fürstenfeldbruck, Germany: (**a**) the gathered rotational sensors in the bunker, (**b**) view of the test field during sensors installation, and (**c**) the data recorded by FOSREMs (type FOS5-01,-02) on the 19 November 2019.

**Figure 12 sensors-24-07003-f012:**
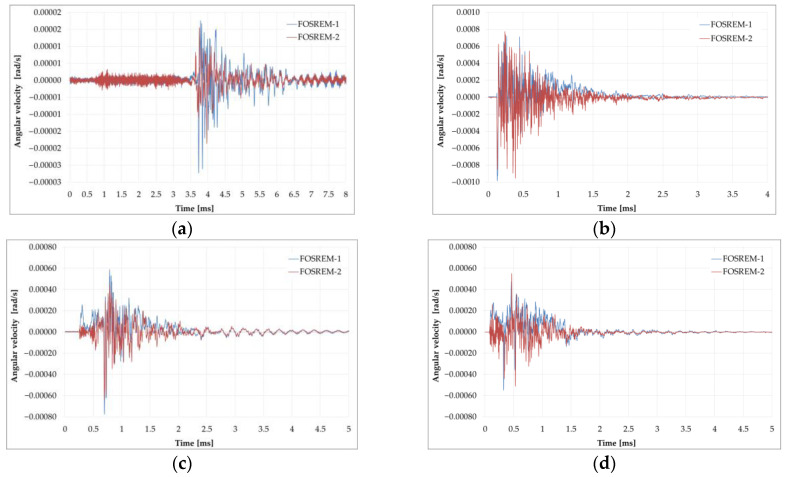
The examples of the torsion (**a**,**b**) and tilt (**c**,**d**) recordings in the frequency range from DC to 10.25 Hz detected by FOSREM-1/-2 from 12/01/2017 to 18/01/2018 at Książ observatory, Poland.

**Table 2 sensors-24-07003-t002:** Parameters of the rotation recordings generated by natural earthquakes. Legend: Y—the year of publication, Ref.—reference, ES—earthquake source, M_w_—magnitude, R—epicentral distance, PGV_h_—peak value of horizontal ground velocity, PGV_v_—peak value of vertical ground velocity, PGω_z,x,y_—peak value of rotational velocity about a particular axis.

Y	Ref.	ES	Sensor	M_w_	R [km]	PGV_h_ [mm/s]	PGV_v_ [mm/s]	PGω_z_ [mrad/s]	PGω_x_ [mrad/s]	PGω_y_ [mrad/s]
1998, 2006	Takeo [59,101]	strike–slip fault, 1997	Systron Donner triaxial gyro sensor	5.7	3.3	290	500	3.3	26	5.9
				5.3	3.3	200	100	8.1	27	30
2009	Takeo [102]	seismic swarm activities at offshore Ito, Japan, 1998	Systron Donner triaxial gyro sensor	5.0	5.6	100	60	3	6	8
				3.6	5.9	6	2	0.2	1	1
				2.4	4.9	6	0.3	0.03	0.2	0.2
2009	Liu et al. [64]	local earthquakes at the HGSD station in Eastern Taiwan	R-1	5.1	51	-	-	0.63	~0.4	~0.3
				2.5–6.63	14.3–260.4	-	-	0.004–0.63	-	-
2010	Brokešová and Málek [103]	earthquake swarm in Western Bohemia, 2008	Rotaphone 3DOF	2.2	4.4	400	-	0.15	-	-
2013	Brokešová and Málek [104]	an earthquake recorded at the station Sergoula, Greece	6 DOF Rotaphone	4.3	5	4.5	9	~0.4	~0.8	~0.7
2016	Yin et al. [105]	215 events at The Garner Valley Downhole Array is in California, 2008–2014	R-1	3.0–7.2	14–207	-	-	0.006–0.453	-	0.004–0.7
2017	Jaroszewicz et al. [106]	local earthquake, Jarocin, Poland	TAPS	3.8	200	-	-	0.005	-	-
			AFORS					0.039		
2018	Ringler et al. [76]	local earthquake	Two SMHD (ATA)	4.2	0.5	22.1	11	1.12/0.85	-	2.11/1.86
		local earthquake		2.8	≤220	-	-	~0.0005	~0.00025	~0.00025
		155 local earthquake		≥2.0	≤220			0.0002–2	0.0002–2	0.0002–2
2020	Wassermann et al. [81]	volcano-tectonic earthquake	BlueSeis-3A	5.3	1.5	2	1	2.4	2.5	2.4
2022	Wassermann et al. [107]	Stromboli volcano, Italy activity	BlueSeis-3A	-	-	<0.01	<0.02	<0.0005	<0.001	<0.001

**Table 3 sensors-24-07003-t003:** Parameters of the recordings associated with teleseismic waves. Legend: Y—year of publication, Ref.—reference, ES—earthquake source mechanism, M_w_—magnitude, R—epicentral distance, PGV_h_—peak value of horizontal ground velocity, PGω_z,x,y_—peak value of rotational velocity about particular axis.

Y	Ref.	ES	Sensor	M_w_	R [km]	PGV_h_ [m/s]	PGω_z_ [nrad/s]	PGω_x_ [nrad/s]	PGω_y_ [nrad/s]
2000	Pancha et al. [88]	New Ireland earthquake, 1999	C-II, G0	7.0	~4700	-	10 (C-II)	5 (G0)	-
		Vanuatu earthquake, 1999	C-II	7.3	~3500		8	-	
2005	Igel et al. [58]	Thrust earthquake Japan	G-ring	8.1	~8830		~35		
2007	Igel et al. [34]	from local event, Germany to Great Andaman earthquake	G-ring	5–9	370–12,700	-	~0.10 –40	-	-
2009	Schreiber et al. [114]	Earthquake Kamachatka, 2006	GEOsensor	7.6	~6500	5197	~10	-	-
		Earthquake Mexico, 2006		5.4	~2000	4646	~5		
		Earthquake California, 2007		3.6	~200	8670	~16		
		Earthquake California, 2007		3.9	~250	14,512	~30		
2011	Lin et al. [87]	Earthquake in Wenchuan Sichuan, China	R-1	7.9	1948	<0.01	1000	10,000	10,000
2012	Belfi et al. [85]	Earthquake in Japan, 2011	G-Pisa	9.0	-	-	~60	-	-
2017	Ross et al. [89]	earthquake Papua New Guinea, 2016	beam rotation sensor BRS	7.9	-	~150 × 10^−6^	-		~30 *
		earthquake Vanuatu, 2016		6.7		~6 × 10^−6^			~2 *
		earthquake New Caledonia, 2016		7.2		~40 × 10^−6^			~10 *
		earthquake north of Ascension Island, 2016		7.1		~15 × 10^−6^			~4.5 *
		earthquake New Zealand, 2016		7.8		~200 × 10^−6^			~60 *
		earthquake of Panguna, Papua New Guinea, 2017		7.9		~150 × 10^−6^			~30 *
2018	Simonelli et al. [90]	Series of earthquakes in Italy, 2016	GINGERino	3.5–5.9	38–77	-	~600–17,000	-	-
2020	Sollberger et al. [73]	Earthquake Gulf of Alaska, 2018	ROMY	7.9	-	-	~6	~8	~4
2021	Igel et al. [86]	Papua New Guinea earthquake, 2019	ROMY	7.6	14,000	-	~5	~9	-
		Turkey earthquake, 2019		5.7	1500		~5	~9	
		Austria earthquake, 2018		3.8	144		~18.9	~18	

* PGω_z,x,y_ [nrad]—ground rotation around the particular axis; in [89], this is referred to as tilt.

**Table 4 sensors-24-07003-t004:** Summary of the main parameters of the large ring laser.

Large Ring Laser	Picture	Year of Installation	Place of Installation	Area [m^2^]	Perimeter [m]	Sensitivity [nrad/s/√Hz]	Long-Term Stability of ΔΩ/Ω_E_
C-I	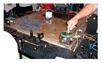 [115,125]	~1992–2011	Cashmere, New Zealand	0.748 (0.85 m square)	3.48	-	-
C-II	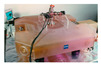 [125,126,127]	1997–2011	Cashmere, New Zealand	1	4	0.146	1 × 10^−6^
G-0	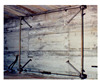 [88,126]	1998–2011	Cashmere, New Zealand	12.25 (3.5 m square)	14	0.0116	5 × 10^−6^
G	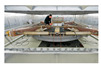 [126,128,129]	2001	Wettzell, Germany	16.0 (4 m square)	16	0.012	3.4 × 10^−8^
UG1	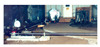 [129,130]	2001–2011	Cashmere, New Zealand	367.5 (17.5 m × 21 m)	77	0.0171	3 × 10^−8^
UG2	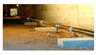 [126,127,129,130]	2004–2011	Cashmere, New Zealand	833.7 (1 m × 39.7 m)	121.4	0.0078	2 × 10^−8^
UG-3	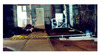 [130,131,132]	2009	Cashmere, New Zealand	367 (7.5 m × 21 m)	77	-	-
GEOsensor	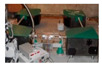 [133]	2005	California, USA	2.56 (1.6 m square)	6.4	0.108	1 × 10^−7^
G-Pisa	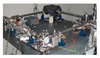 [109,112,134,135]	2008	Pisa, Italy	1.96 (0.9 m × 0.9 m to 1.4 m × 1.4 m)	5.4	~ 1	2 × 10^−5^
GINGERino	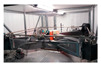 [136]	2014	Pisa, Italy	12.96	14.	0.1	~10^−6^
GP-2	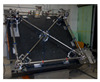 [94,137,138]	2014	Pisa, Italy	2.56	6.4	2	~6 × 10^−5^
ROMY	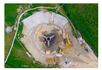 [86,139]	2016	Fürstenfeldbruck, Germany	72 (3 vertical)50 (horizontal)	3630	0.08–0.1	5 × 10^−5^

**Table 8 sensors-24-07003-t008:** Brief presentation of Rotaphone-CY record parameters obtained during explosions based on [163]; Legend: A—amount of explosive, D—distance from Rotaphones, PGω_z,x,y_—maximum absolute values of rotational components around the particular axis, PGV_x,y,z_—maximum absolute values of translational components along the particular axis, F—prevailing frequency, calculated as the instantaneous frequency of maximum rotation rate.

Event Parameters			PGω [µrad/s]			PGV [µm/s]			F [Hz]		
No.	A [g]	D [m]	PGω_z_	PGω_x_	PGω_y_	PGV_z_	PGV_x_	PGV_y_	*Z*	*X*	*Y*
1.	150	220	2.4	2.6	3.0	22.9	16.3	15.2	15.8	11.7	12.6
2.	500	52	21.5	13.4	145.7	418.6	191.9	55.2	16	15.3	16.6
3.	1500	452	1.5	1.4	1.7	17.4	14	17.4	10.3	8.8	7.3
4.	1500	676	1.6	1	1.4	10.4	9.6	11.6	10.7	9.6	6.3
5.	1500	1020	0.8	1.1	0.8	5.2	6.4	11.4	9.4	10.1	6.8

**Table 10 sensors-24-07003-t010:** List of parameters of exemplary rotational events recorded by FOSREM-1 and -2 in the period 12 January 2017–25 January 2018 associated with mining activity.

Torsion				Tilt		
FOSREM-1/-2	Date	Event Start	Max. Signal Amplitude [µrad/s]	Date	Event Start	Max. Signal Amplitude [µrad/s]
FOSREM-1/-2	8 January 2018	08:09:50	3.89/1.81	22 September 2017	06:54:07	102/101
FOSREM-1/-2	14 December 2017	08:11:48	5.45/2.84	12 December 2017	08:54:27	22.8/15.3
FOSREM-1/-2	28 November 2017	09:16:24	19.9/10.1	25 January 2018	09:44:47	52.1/42.0
FOSREM-1/-2	25 January 2018	09:40:05	4.93/2.97	3 February 2018	10:14:23	240/219
FOSREM-1/-2	1 December 2017	10:04:21	8.25/4.17	1 January 2018	10:48:39	101/117
FOSREM-1/-2	1 December 2017	10:05:55	16.7/10.3	14 December 2017	10:51:04	27.4/46.5
FOSREM-1/-2	28 November 2017	10:36:11	1.86/1.61	29 August 2017	11:02:53	36.6/22.0
FOSREM-1/-2	28 November 2017	10:36:54	1.58/ 1.01	6 December 2017	11:04:58	17.3/12.0
FOSREM-1/-2	14 December 2017	10:56:31	1.95/ 1.27	13 December 2017	11:15:56	2.53/2.79
FOSREM-1/-2	6 December 2017	10:59:29	5.34/3.24	8 December 2017	13:01:23	96.1/77.5
FOSREM-1/-2	5 October 2017	11:26:39	20.0/10.0	13 December 2017	17:11:46	35.5/29.0
FOSREM-1/-2	11 December 2017	13:49:25	9.83/5.15	13 December 2017	18:01:32	35.6/77.1
FOSREM-1/-2	13 December 2017	14:51:07	1.65/1.32	13 December 2017	18:06:43	34.7/62.0
FOSREM-1/-2	20 October 2017	16:33:43	7.34/6.28	13 December 2017	18:07:11	58.1/85.6
FOSREM-1/-2	13 December 2017	17:01:38	31.5/15.5	13 December 2017	18:11:45	55.3/97.8
FOSREM-1/-2	13 December 2017	18:25:56	1.77/1.09	13 December 2017	19:16:13	1.67/1.99

## Data Availability

The data supporting this study’s findings are available from the corresponding author, [A.T.K.], upon reasonable request. Some of the data are shared on the website: https://fosrem.eu, accessed on 18 October 2024.

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
