# Peer review of "A Review of Rotational Seismology Area of Interest from a Recording and Rotational Sensors Point of View"

_sensors, 2024, doi:10.3390/s24217003_

Round 1
Reviewer 1 Report
Comments and Suggestions for Authors
The paper provides a comprehensive review of the field of rotational seismology, highlighting the theoretical foundations and recent advancements in recording rotational motions during seismic events. It covers various research areas, such as earthquake-induced rotational motion, teleseismic waves, artificial explosions, and mining activities. Special focus is given to the Fiber-Optic System for Rotational Events & Phenomena Monitoring (FOSREM), a sensor developed to measure rotational movements, capable of detecting a wide range of rotation rates. The authors emphasize the growing importance of rotational seismology in improving the understanding of seismic wave behavior and structural health monitoring.
The paper offers a well-organized and detailed review of rotational seismology, presenting both theoretical and practical aspects. It successfully compiles relevant research and highlights the importance of integrating rotational measurements with translational ones to enhance the accuracy of seismic analyses. The inclusion of FOSREM's capabilities is a valuable contribution to understanding modern measurement tools. However, the paper could benefit from more discussion on the limitations of current sensor technologies and future directions in the field. The following are other comments.
1. The paper effectively explains the relevance of rotational components in seismic events, but some sections could be simplified for broader audiences.
2. Figures, such as those showing gravestones deformed by seismic rotations, are useful, but additional diagrams explaining sensor functionalities (like FOSREM) would further enhance clarity.
3. While FOSREM's strengths are well-documented, the paper does not sufficiently address the limitations and challenges of these sensors in various environments, especially regarding thermal stability and sensor noise.
4. The theoretical section provides rigorous equations, but some parts are dense. A brief explanation of their practical implications for seismologists would be helpful.
5. More focus could be placed on the potential integration of rotational seismology with emerging technologies, such as machine learning, for data analysis and prediction.
6. The paper briefly touches on future prospects but could explore in more depth the possible innovations and interdisciplinary applications of rotational seismology in fields like structural engineering and environmental monitoring.
Reviewer 2 Report
Comments and Suggestions for Authors
The authors examine the role of rotational seismology, considering different areas of interest, as well as measuring devices. They provide a comparative review of numerical investigations of rotational effects, rotation measured during earthquakes, teleseismic waves, mining activities. They include a discussion of the importance of rotational seismology and the need to acquire data in this area. The paper is well written, and the results of numerical investigations make the investigation of the research seem promising. The reviewer believes that the paper is good enough for publication in a journal. A few minor comments are listed below.
- In lines 283-298, it would be helpful to the reader to indicate how these seismic parameters are selected, e.g. by adding references.
- Could the authors elaborate on the statistical properties (e.g., RMSE) for the peak amplitude vs. epicentral distance and magnitude in Figure 3?
- In line 1049, the authors might want to elaborate on the reference for the “recent research” showing the contribution of the rotational component of the vibrations.
- Double periods are used in line 1141.
